# *Mentha pulegium* L. (Pennyroyal, Lamiaceae) Extracts Impose Abortion or Fetal-Mediated Toxicity in Pregnant Rats; Evidenced by the Modulation of Pregnancy Hormones, MiR-520, MiR-146a, TIMP-1 and MMP-9 Protein Expressions, Inflammatory State, Certain Related Signaling Pathways, and Metabolite Profiling via UPLC-ESI-TOF-MS

**DOI:** 10.3390/toxins14050347

**Published:** 2022-05-16

**Authors:** Amira A. El-Gazar, Ayat M. Emad, Ghada M. Ragab, Dalia M. Rasheed

**Affiliations:** 1Pharmacology and Toxicological Department, Faculty of Pharmacy, October 6 University, Sixth of October City 12585, Egypt; amira.ahmed@o6u.edu.eg; 2Pharmacognosy Department, Faculty of Pharmacy, October 6 University, Sixth of October City 12585, Egypt; ayatEmad@o6u.edu.eg; 3Pharmacology and Toxicological Department, Faculty of Pharmacy, Misr University for Science & Technology (MUST), Giza 12585, Egypt; ghada.ragab@must.edu.eg

**Keywords:** *Mentha pulegium* (pennyroyal), abortion, UPLC metabolite profiling, IUGR, micro-RNA, MMP-9

## Abstract

Pregnant women usually turn to natural products to relieve pregnancy-related ailments which might pose health risks. *Mentha pulegium* L. (MP, Lamiaceae) is a common insect repellent, and the present work validates its abortifacient capacity, targeting morphological anomalies, biological, and behavioral consequences, compared to misoprostol. The study also includes untargeted metabolite profiling of MP extract and fractions thereof *viz.* methylene chloride (MecH), ethyl acetate (EtOAc), butanol (But), and the remaining liquor (Rem. Aq.) by UPLC-ESI-MS-TOF, to unravel the constituents provoking abortion. Administration of MP extract/fractions, for three days starting from day 15th of gestation, affected fetal development by disrupting the uterine and placental tissues, or even caused pregnancy termination. These effects also entailed biochemical changes where they decreased progesterone and increased estradiol serum levels, modulated placental gene expressions of both MiR-(146a and 520), decreased uterine MMP-9, and up-regulated TIMP-1 protein expression, and empathized inflammatory responses (TNF-α, IL-1β). In addition, these alterations affected the brain's GFAP, BDNF, and 5-HT content and some of the behavioral parameters escorted by the open field test. All these incidences were also perceived in the misoprostol-treated group. A total of 128 metabolites were identified in the alcoholic extract of MP, including hydroxycinnamates, flavonoid conjugates, quinones, iridoids, and terpenes. MP extract was successful in terminating the pregnancy with minimal behavioral abnormalities and low toxicity margins.

## 1. Introduction

Abortifacients are compounds that induce miscarriage or abortion, and the herbs used for such purposes, either intentionally or by misconception may pose toxicity risks to the administering mother. The use of herbs and plant extracts to intervene with pregnancy and enforce menses is one ancient folk practice that entails complex social, legal, ethical, and health aspects [1].

Plants of genus Mentha (Lamiaceae) are aromatic, medicinal, and ornamental herbaceous plants with a wide distribution all over the world as they can flourish in diverse habitats. Varieties of Mentha are extensively used in food, cosmetics, and in aroma and in complementary therapies, primarily for the treatment of gastrointestinal complaints, such as indigestion, flatulence, spasm, nausea, irritable bowel syndrome, and ulcerative colitis [2]. *Mentha pulegium* L. (Pennyroyal) is an aromatic perennial herb of the mint family with a strong spearmint-like odor. Earlier it was widely cultivated and used traditionally as an insect repellent, food preservative, emmenagogue, and to relieve menstruation ailments [3]. However, its distribution and use have declined remarkably in recent years, because of pennyroyal toxicity claims to humans, especially regarding its pulegone rich essential oil [4,5]. One of the earlier traditional practices pertained to pennyroyal consumption was to terminate undesired pregnancies or to expel dead fetuses [6]. In fact, the use of pennyroyal infusion as a natural abortifacient has been repeatedly reported, even Carl Linnaeus, listed pennyroyal as an abortifacient in his 1749 Materia medica [1,6,7]. However, the mechanism of pregnancy termination/ abortion and correlating this action to pennyroyal’s metabolic profile has yet to be investigated. According to the phytochemical investigations, the chemical composition of pennyroyal can be divided into two major groups; water-soluble phytoconstituents viz. phenolic acids, flavonoids and dihydrochalcone glycosides in addition to lipophilic fatty acids and terpene derivatives [8]. Flavonoid compounds previously reported in pennyroyal include derivatives of quercetin, isorhamnetin, naringenin, and gallocatechins, whereas hydroxycinnamic acid derivatives include rosmarinic and salvianolic acid conjugates signifying the Lamiaceae members [9].

Misoprostol is currently the drug of choice for induction of medical abortion as a single regimen in many regions where mifepristone is not indexed/unreachable or expensive, in addition to the convince in its use/administration [10,11,12]. Misoprostol is a prostaglandin E1 analogue with a wide range of applications, particularly in obstetrics and gynecology [13]. Misoprostol induces abortion through raising uterine tone and contractions in pregnant women [13,14]. However, misoprostol unsuccessful attempt may be associated with fetal deformity, and congenital abnormalities, and the reasons surrounding these effects are still vague and requires more investigation [15,16].

The scientific progress towards characterizing the role of natural product involvement in jeopardizing pregnancy and unrevealing the pharmacological action and psychological consequences is still very limited. This research can be regarded as a decisive endeavor for settling the potential abortifacient capacity of MP extract by unraveling the underlined mechanistic pathway and analyzing the incidences associated with its intended and/or misuse in comparison to misoprostol. Considering that the failure of early pregnancy termination, through folk practice or medical procedures, is associated with increased incidence of a variety of disorders inflected on the mother and fetus, hence tracking the congenital abnormalities, teratogenicity, biological, behavioral, and pathological consequences is momentous.

## 2. Results

### 2.1. Dose Selection

#### 2.1.1. Abortifacient and Biochemical Effects of Different Doses of *Mentha pulegium* L. Body Weight-Related

Administration of *Mentha pulegium* L. extract in a dose of 250 and 500 mg/kg resulted in complete abortion of two of the six pregnant rats relative to the effect of the standard drug misoprostol (three of the six rats), whereas 125 mg/kg revealed less effectiveness as it did not show a complete abortion among its group. However, the 125 mg/kg dose resulted in a reduction in progesterone levels and showed signs of toxicity viz. asymmetric distribution of fetuses in the two uterine horns, absorption points, hematoma, and absence of boundaries between fetal balls; presented in Table 1, Figure 1 (panel I) as red brackets, green arrows, star, and yellow arc, respectively.

#### 2.1.2. Histopathological Effects of the Different Doses of *Mentha pulegium* L. on Uterine Tissues

The paraffin sections as imaged in Figure 1(IIB) pregnant uterine tissue revealed more thickening of the uterine wall with mild vacuolization of lining epithelium (black arrow), as well as mild hyperplasia of uterine gland (yellow arrow) and many dilated and congested blood vessels of intermuscular tunica vascular (star) in pregnant rats as compared tox Figure 1(IIA) non-pregnant rats that demonstrated the apparent intact uterine wall with almost intact lining epithelium with scattered apoptotic cell bodies (black arrow) moderate submucosal mononuclear cells infiltrates (red arrow) alternated with normal connective tissue and intact myometrium, as well as normal intermuscular blood vessels (star). Furthermore, treatment with different doses of the MP extract presented an order in efficacy/power, that was augmented by increasing the extract dose, where Figure 1(IID) 125 mg/kg of MP showed mild endometritis with mild neutrophilic infiltrates (red arrow), as well as mild congested uterine vasculatures (star). The Figure 1(IIE) MP 250 mg/kg dose demonstrated more apparent endometritis with moderate vacuolar changes of lining uterine epithelium (black arrow). Eventually, Figure 1(IIF) MP 500 mg/kg dose showed the worst histo-findings where significant hyperplasia of lining epithelium (black arrow) with many intra epithelial inflammatory cells infiltrates with milder infiltrates (red arrow) in submucosal layers and mild congested uterine BVs (star) and abundant neutrophilic infiltrates (red arrow) with mild congested uterine BVs (star) were observed as compared to medically aborted rats using Figure 1(IIC) misoprostol that manifested significant atrophy of lining epithelium (black arrow), cystic dilatation of many endometrial glands with abundant intraluminal desquamated lining epithelium (yellow arrow) accompanied with sever endometritis and lymphocytic infiltrates (red arrow).

#### 2.1.3. Acute Toxicity Test

No deaths were observed. However, at the dose level of 500 mg/kg, signs of toxicity on uterine tissue were observed as mentioned previously.

### 2.2. Identification of the Abortifacient Mechanism of MP Extract and Fractions Thereof

#### 2.2.1. Abortifacient Effects/Activities of 250 mg/kg and Different Fractions of *Mentha pulegium* L.

Rats treated with MP extract or misoprostol behaved similarly in terms of the number of completely aborted rats, the change in weight of pregnant rats/mothers and the number and weights of remaining fetuses relative to positive control mothers and healthy fetuses (Table 2, Figure 2I). On the other hand, the different fractions did not show complete abortion, but a significant reduction in fetal weight was observed, and was more pronounced with the but. and remaining aqueous fractions showing the maximum weight reduction in mothers on 18th day compared to the 15th day of gestation and the highest intrauterine growth retardation (IUGR) detected by fetal weight. Additionally, the MP and different fractions generated a different pattern of toxicity on uterine tissue, as shown in Figure 2(II), such as hematoma, intensive bleeding, asymmetrical distribution of fetuses in the two horns of uterus, complete absence of fetus on one uterine horn, resorption sites, absence of boundaries between fetal balls, and small fetal balls. Panel III presents IUGR associated with Figure 2(IIIb) misoprostol Figure 2(IIIc) MP extract and its fractions administration, as implied by their weights and lengths, morphologic malformations, hematoma, transparency of skin, and intensive bleeding as compared to those of fetuses in Figure 2(IIIa) control group. We omitted the fetal lengths measurements although there was significant difference between groups as high elasticity and flexibility of fetus may be a misleading during measurements and increase error.

#### 2.2.2. Effect of *Mentha pulegium* L. on the Serum Levels of Progesterone and Estradiol

As shown in Figure 3I, pregnant female rats treated with misoprostol (100 ug/kg) and 250 mg/kg of MP extract both, had a marked decline in serum Figure 3(IA) progesterone levels and this effect reached Figure 3(IB) estradiol levels which was increased as compared to untreated pregnant female rats. It is worth mentioning that both MP extract and the standard drug, misoprostol produced similar values, where the progesterone levels were 11.17 ± 1.6 and 9.70 ± 2.0 ng/mL and the estradiol levels were 76.30 ± 3.0 and 72.55 ± 7.0 pg/mL, respectively. Meanwhile, the different fractions of MP extract also exhibited an abortifacient effect by the observed decline in serum progesterone levels. Additionally, the butanol and the remaining aqueous fractions effectively increased the levels of estradiol as compared to healthy pregnant one and those treated with total extract.

#### 2.2.3. Effect of *Mentha pulegium* L. on Placental Protein Expressions of MiR-520 and MiR-146a

Figure 3II illustrates the disturbance in micro-RNA (MiR) expression in all the pregnant rats after misoprostol or MP (extract; 250 mg/kg) treatment where both increased Figure 3(IIA) MiR-520 and decreased Figure 3(IIB) MiR-146a protein expression as compared to the pregnant ones. Furthermore, these perturbations in both MiRs’ expression were also figured in groups of the different fractions, knowing that Rem.aq. showed the highest increase in MiR-520 among fractions as compared to MP values, however all the fractions showed almost the same inhibitory effect as the total extract on MiR-146a.

#### 2.2.4. Effect of *Mentha pulegium* L. on Uterine Protein Expressions of MMP-9 and TIMP-1

As revealed in Figure 3III, a marked elevation in Figure 3(IIIA) TIMP-1 that in sequence decreased Figure 3(IIIB) MMP-9 protein expression was detected on treatment with misoprostol or MP extract as compared to untreated pregnant female. Furthermore, medically aborted rats by our chosen standard showed a higher effect on the uterine TIMP-1 and MMP-9 expression as compared to values of MP extract. Regarding the fractions, they showed similar activity to total MP extract on TIMP-1 and MMP-9 expression with the MecH fraction exhibiting the highest effect on TIMP-1 and MMP-9 expressions, whereas the Rem. aq. fraction was more dominant on TIMP-1 expression.

#### 2.2.5. Effect of *Mentha pulegium* L. on Serum Inflammatory Markers

Figure 4I depicts the effect of MP extract administration on the levels of inflammatory mediators, Figure 4(IA) tumor necrosis factor alfa (TNF-α) and Figure 4(IB) interlukin-1 beta (IL-1β) which were emphatically spiked after its administration for three days and even exceeding them misoprostol effects on TNF-α levels (29.80% and 56.17% of control pregnant rats, respectively). Furthermore, the inflammatory activity of all the fractions under investigation were confirmed by the elevated inflammatory markers as compared to MP extract. Nevertheless, the Rem. Aq. Fraction recorded the highest increase in TNF-α among all the animals to reach 76.67% of normal pregnant ones. Concerning IL-1β, the But. And Rem. Aq. Fractions gave no significant difference compared to MP extract which marks a more significant effect in accordance with their employed doses.

#### 2.2.6. Effect of *Mentha pulegium* L. on Behavioral Changes in Open Field Test

In the open field test (Figure 4II), female rats behaved differently after misoprostol, MP, and different fraction administration, indicating a variety of behavioral abnormalities associated with medically induced abortion. The ones treated with misoprostol and EtOAc fraction showed immobilization for Figure 4(IIA) 3 and 5 s., respectively, as represented by increase in the time from putting the animal into the middle of the device until its first movement (latency time), reflecting a defect in motor function by decreasing locomotor activity as compared to the untreated pregnant female. However, the MP extract and the other fractions *viz.* MecH, But. And Rem.aq. showed no latency lag compared to the control rats. The increased Figure 4(IIA) latency time in both misoprostol and EtOAc fraction was reflected on and extended to Figure 4(IIB) ambulation and Figure 4(IIC) rearing frequencies, where both showed reduced records relevant to MP extract and other fractions except the Rem.aq. fraction. This exceptional increase in rearing and ambulation frequency occurring by the Rem.aq. fraction treatment, may reflect anxiety and stress. The MP extract and its fractions reduced Figure 4(IID) grooming frequency, except for the Rem.aq. fraction treated group, which showed a significant increase in the number of times of rapid cleaning movements of the forelegs towards the face and/or the body (licking the fur, washing the face, or scratching behavior).

#### 2.2.7. Effect of *Mentha pulegium* L. on Cortical GFAP, BDNF, and 5HT-3 Levels

Misoprostol treatment affected the brain where a decreased in cortical contents of Figure 4(IIIA) GFAP was detected. Surprisingly, MP treatment on day 15th of gestation and for three days saved GFAP levels in cortical brain tissue and showed comparable levels of GFAP to the positive control. Additionally, all the fractions exerted the same effect of MP extract except for the Rem.aq. fraction which demonstrated a marked decline in GFAP levels as compared to MP extract results. However, all the rats when compared to the positive control demonstrated a significant reduction in cortical Figure 4(IIIB) BDNF levels after misoprostol and MP extract and/or fractions treatments; where all the fractions showed similar patterns with no statistical difference when compared to their corresponding extract except for But. The fraction did not affect BDNF levels showing the same values of control pregnant rat. It is worth mentioning that the decline in BDNF content by misoprostol treated group reached almost half of contents of the control group. Ultimately, Figure 4(IIIC) 5HT cortical contents were massively increased in misoprostol and MP extract treated rats as compared to the positive control. However, all the fractions did not affect serotonin levels except EtOAc fraction which caused a significant increase in its levels as compared to the ones with normal gestation. These results were recapitulated in Figure 4III.

### 2.3. Metabolite Profiling of Mentha pulegium *L.* (Pennyroyal) Extract Using UPLC-ESI-TOF-MS

UPLC–ESI–TOF-MS was employed for metabolome characterization of *Mentha pulegium* L. (MP, pennyroyal) and fractions thereof in a holistic, untargeted manner and correlate its phytochemical configuration to the anticipated abortifacient effect. The analytical procedure adopted herein warranted an adequate elution of the analytes within 30 min in an order of decreased polarity (Figure 5). A total of 128 metabolites were detected in the 70% ethanol extract of pennyroyal, classified according to their phytochemical structures into hydroxycinnamates, several flavonoid conjugates, quinones, iridoids, and terpenes. A Tandem mass experiment was useful in distinguishing between O/C attached flavonoid and phenolic conjugates. Generally, the *O*-glycosyl attachment can be readily identified in MS^2^ spectra by the neutral losses of 162, 146, and 132 amu indicative of hexose, deoxyhexose or pentose moieties, respectively, whereas the fragmentation pattern of the *C*-glycosyl conjugates shows major fragment ions resulting from neutral losses of 90 and 120 amu of pentose and hexose sugars, respectively [17].

Details for detection and assigning major metabolites, listed in Table 3 and depicted in Figure 5 and Figure 6, are explained in the following section.

#### 2.3.1. Hydroxycinnamic Acids and Derivatives

Fourteen hydroxycinnamic acid conjugates were detectable in MP metabolite profile, half of which were eluted first and occupied the early section of the chromatogram at a retention time range (t_R_) of 1.31–5.17 min. Caffeoyl-*O*-glucuronide assigned for peak 1 with molecular ion *m*/*z* 355.0852 [M − H]^−^ (t_R_ = 1.31 min, and predicted chemical formula [C_15_H_15_O_10_]^−^), exhibited two daughter ions; *m*/*z* 193.0739 and *m*/*z* 161.0428 in MS^2^ spectra, representative of glucuronyl and caffeoyl fragments, respectively (Appendix A). Sinapoyl-*O*-hexoside, ascribed for molecular ion *m*/*z* 387.1408 [M + H]^+^ (peak 5, t_R_ = 1.87 min, [C_17_H_23_O_10_]^+^), generated a daughter ion; *m*/*z* 225.0784 with predicted chemical formula [C_11_H_13_O_5_]^+^, corresponding to a sinapoyl moiety and the mass difference of (162 amu) between the two ions signifies the loss of the hexose moiety (Appendix A). Caffeic acid dimers and oligomers, termed “depsides” *viz.* salvianolic and rosmarinic acids, are discriminative phytochemical markers of family Lamiaceae [18]. Five caffeoyl dimer metabolites were tentatively identified in MP extract; peaks # 4, 8, 9, 10, and 11 as caffeic acid dimer, salvianolic acid K, caffeoyl-sinapoylquinic acid, salvianolic acid F and rosmarinic acid, respectively. Rosmarinic acid, the ester of 3,4-dihydroxyphenyllactic acid (danshensu, or salvianic acid A) and caffeic acid, was ascribed to molecular ion *m*/*z* 361.0907 [M + H]^+^ (peak 11, t_R_ = 12.55 min, [C_18_H_17_O_8_]^+^), where the MS^2^ examination showed a fragment at *m*/*z* 197.0104 ([M + H – C_9_H_10_O_5_]^+^) signifies the danshensu moiety, and the neutral loss of 164 amu signifies the caffeoyl group (Appendix A). Peak 8 at *m*/*z* 557.1252 [M + H]^+^, (t_R_ = 7.26 min, [C_27_H_25_O_13_]^+^) gave fragments at *m*/*z* 511.0066 ([M + H – CO_2_]^+^) and 363.1261 ([M + H – C_9_H_6_O_5_]^+^, considered for the loss of carboxylic group and danshensu, respectively, and the compound was identified as salvianolic acid K (Appendix A). Salvianolic acid F, another depside of caffeic acid was annotated to the ion at *m*/*z* 315.0836 [M + H]^+^ (peak 10, t_R_ = 8.98 min, [C_17_H_15_O_6_]^+^), revealed a fragment ion at *m*/*z* 181.0863 with the chemical formula [C_9_H_9_O_4_]^+^, signifying the caffeic acid moiety (Appendix A). Although other salvianolic acids have been reported in pennyroyal, to our knowledge, salvianolic acids K and F are reported herein for the first time in Egyptian MP, in addition to caffeoyl-sinapoylquinic acid (Appendix A).

#### 2.3.2. Flavones and Derivatives

The flavonoid configuration of MP extract was dominated by 18 flavone compounds detected with varying abundances and eluted in the order of 10 flavone conjugates (t_R_ = 4.86–10.32 min). Followed by 8 flavone aglycones (t_R_ = 10.42–16.88 min) (Figure 5). Apigenin 6,8-di-*C*-hexoside (vicenin 2) was assigned to the ion detected at *m*/*z* 593.1489 [M – H]^−^ (peak 15, t_R_ = 4.87 min, [C_27_H_29_O_15_]^−^) based on the fragment ions at *m*/*z* 353.0645, 383.0711 amu for [Apg + 83/Apg + 113]^−^ typifying a di-*C*-glycoside of apigenin [19], (Appendix A). Vicenin 2 was reported in *Origanum* species of Lamiaceae, but not in pennyroyal [9].

Three glucuronide conjugates of penduletin, luteolin and apigenin were assigned to molecular ions at *m*/*z* 521.1812, 463.0899, and 447.0827 amu, peaks 16, 21, and 22, respectively (Appendix A), where the MS^2^ data showed the fragment ions respective to the aglycones at *m*/*z* 345.1507, 287.0550, and 271.0593 signifying the loss of a glucuronide moiety [M + H − 176]^+^. The flavone glycoside diosmin (diosmetin 7-*O*-rutinoside), which has been repeatedly reported in several members of the genus Mentha [9] was ascribed to the ion at *m*/*z* 609.1782 [M + H]^+^, (peak 20, t_R_ = 7.79 min, [C_28_H_33_O_15_]^+^) by the abundance of the daughter ions at *m*/*z* 463.1208 and 301.0695 amu, indicative of the cleavage of the deoxyhexose (146 amu) followed by the hexose moieties (162 amu), respectively (Appendix A). A molecular ion was observed with high abundance at *m*/*z* 327.2178 [M − H]^−^ (peak 17, t_R_ = 7.15 min, [C_18_H_15_O_6_]^−^), was tentatively identified as the flavone salvigenin after generating a daughter peak at *m*/*z* 190.9540 amu denoting the deprotonated dimethoxy substituted moiety left after C-ring cleavage of the aglycone by Retro-Diels–Alder reactions (Appendix A) [20]. Similarly, acacetin was annotated to the molecular ion at *m*/*z* 285.0753 [M + H]^+^ (peak 29, t_R_ = 14.23 min, [C_16_H_13_O_5_]^+^), after the aglycone cleavage along C-ring to give the fragment ion at *m*/*z* 153.0169 amu (Appendix A). Acacetin-*O*-rutinoside (linarin) was readily assigned to the ion at *m*/*z* 593.1879 [M + H]^+^ (peak 23, t_R_ = 9.35 min, [C_28_H_33_O_14_]^+^) which yielded the fragment ions at *m*/*z* 447.1286 after loss of the deoxyhexose (146 amu) followed by the fragment ions at *m*/*z* 285.0767 representing the aglycone after the loss of a hexose moieties (162 amu) (Appendix A).

#### 2.3.3. Flavanones and Flavanol Derivatives

Seven flavanones and flavanol derivatives were detected in the 70% ethanol metabolite profile of pennyroyal. Naringenin-7-*O*-glucuronide was assigned to the molecular ion at *m*/*z* 449.1112 [M + H]^+^ (peak 33, t_R_ = 5.07 min, [C_20_H_21_O_11_]^+^), whereas eriodictyol-7-*O*-glucuronide was assigned to the molecular ion at *m*/*z* 463.0990 [M − H]^−^ (peak 36, t_R_ = 8.16 min, [C_21_H_19_O_12_]^−^) by the observed ions in MS^2^ spectra at *m*/*z* [273.0965]^+^ and [286.9356]^−^, respectively, implying the neutral loss of a 176 amu for the glucuronide moiety (Appendix A). A major peak (39), eluded at t_R_ = 21.51 min in the positive ionization mode, produced an intense molecular ion at *m*/*z* 611.2864 that exhibited a predicted chemical formula [C_28_H_35_O_15_]^+^. The metabolite was annotated hesperetin 7-*O*-rutinoside (hesperidin) as reported previously in pennyroyal [21] and confirmed by the fragment ion at *m*/*z* [465.2265]^+^ implying the loss of a deoxyhexose portion (146 amu) (Appendix A).

#### 2.3.4. Prenyl Flavanones

Mass spectrometry-based detection is a powerful tool for discovering minor secondary metabolites of potential medicinal merit in plant extracts [22]. The analytical procedure adopted in this study succeeded in tentative identification of seven prenylated flavones in MP extract for the first time. A molecular ion of low abundance at *m*/*z* 353.2329 [M − H]^−^ (peak 41, t_R_ = 7.24 min, [C_20_H_17_O_6_]^−^) was assigned prenyl kaempferol after the neutral loss of 68 amu (isoprenyl group) to yield a fragment ion at *m/z* [285.1418]^−^ with the predicted chemical formula of kaempferol [C_15_H_9_O_6_]^−^ (Appendix A). Similarly, another molecular ion of low abundance at *m*/*z* 339.216 [M − H]^−^ (peak 45, t_R_ = 14.99 min, [C_20_H_19_O_5_]^−^) was annotated prenyl naringenin, generating the daughter ion at *m*/*z* [271.2282]^−^ with the predicted chemical formula of naringenin [C_15_H_11_O_5_]^−^ after the loss of 68 amu (Appendix A).

#### 2.3.5. Biflavanoids

Amentoflavone has been reported previously in horse mint (*Mentha longifolia*) [23] and sweet marjoram (*Origanum majorana*) [24] of family Lamiaceae. In this study, five biflavonoids have been tentatively identified in the metabolome profile of MP for the first time. Di-*O*-methylamentoflavone (ginkgetin) was assigned to the molecular ion at *m*/*z* 567.1288 [M + H]^+^ (peak 50, t_R_ = 18.88 min, [C_32_H_23_O_10_]^+^) which gave in MS^2^ spectra a base peak at *m*/*z* [297.0760]^+^ typifying the di-*O*-methyl-apigenin monomer of the biflavonoid molecule, and a fragment ion at *m*/*z* [165.0206]^+^ after further cleavage along its C-ring (Appendix A). The methyl derivative of ginkgetin, sciadopitysin (amentoflavone-7,4′,4‴-trimethyl ether), was allocated to the molecular ion at *m*/*z* 581.1424 [M + H]^+^ (peak 51, t_R_ = 21.59 min, [C_33_H_25_O_10_]^+^) after generating the same base peak at *m*/*z* [297.0760]^+^ in MS^2^ spectra (Appendix A).

#### 2.3.6. Terpenes

Members of the Lamiaceae are known to accumulate a wide variety of mono, diterpenes, or even triterpenes of significant medicinal and industrial value. The Dictionary of Natural Products listed more than 2000 diterpene constituents in family Lamiaceae [25]. Labdane core diterpenes feature Lamiaceae characteristic marker metabolites and in accordance, abienol (peak 70, t_R_ = 13.53 min, *m*/*z* 291.0687 [M + H]^+^, [C_20_H_35_O]^+^) and coleonol (peak 75, t_R_ = 18.94 min, *m*/*z* 409.2562 [M – H]^−^, [C_22_H_33_O_7_]^−^) were tentatively identified in MP profile (Appendix A). The phenolic diterpenes, carnosic acid (peak 69, t_R_ = 13.33 min, *m*/*z* 333.2042 [M + H]^+^, [C_20_H_29_O_4_]^+^), and carnosol (peak 71, t_R_ = 13.88 min, *m*/*z* 331.1913 [M + H]^+^, [C_20_H_27_O_4_]^+^) are known active secondary metabolites which have been reported in rosemary (Lamiaceae) [26] (Appendix A). Two abietane diterpenoids, pachyphyllone (peak 76, t_R_ = 19.01 min, *m*/*z* 317.2122 [M + H]^+^, [C_20_H_29_O_3_]^+^) had a fragmentation pattern matching that reported in [27] (Appendix A), in addition to dihydrotanshinone I, (peak 83, t_R_ = 24.91 min, *m*/*z* 279.1027 [M + H]^+^, [C_18_H_15_O_3_]^+^) another characteristic diterpenes of Lamiaceae characterized by a base peak at *m*/*z* [149.0241] ^+^, for benzofuran-4,5-dione in MS^2^ spectra [28] (Appendix A). Asiatic acid (peak 85, t_R_ = 17.22 min., *m*/*z* 489.3607 [M + H]^+^, [C_30_H_48_O_5_]^+^) exhibited a fragmentation pattern matching that reported in [29] (Appendix A). Ursolic acid (peak 92, t_R_ = 23.25 min., *m*/*z* 457.3684 [M + H]^+^, [C_30_H_48_O_3_]^+^) was assigned after generating fragmentation ions in MS2 spectra at *m*/*z* [439.3539]^+^ and [411.3628]^+^ for dehydration and decarboxylation of the parent ion, respectively. According to the data, the fragmentation ions at *m*/*z* [203.1787]^+^ and [191.1794]^+^ are Retro-Diels-Alder reaction products for Δ 12-unsaturated pentacyclic triterpenes [30] (Appendix A).

## 3. Discussion

Pregnant women usually turn to self-medication, relying on herbal and natural products to relieve pregnancy related ailments, or even for pregnancy termination, regardless of the lack of evidence concerning the safety profile of these products. This study targets the biological, behavioral, pathological, and toxicological consequences prior to administering the 70% ethanol extract of *Mentha pulegium* L. (MP) in pregnant female rats. This aim was executed over two parts; the first part includes probing the abortifacient potential using different doses of MP 70% ethanol extract to specify the effective abortifacient dose. Designating the abortifacient effect will rely on assessing some parameters, including percentage of abortion per group, number of resorbed fetuses, and further correlation to serum progesterone levels for confirmation of the results. On the 18th day of gestation, animals were sacrificed, and the uteri were examined. After specifying the dose of 250 mg/kg of MP extract as an abortifacient dose comparable to the standard misoprostol, the study attempts, in the second part, to unravel the mechanistic pathway underlying the abortifacient action. Additionally, the fractions of the 70% ethanol extract of MP, prepared as described in Section 5.1, were also similarly examined to trace the phytochemical constituents responsible for this effect.

Maternal exposure to MP (extract and/or fractions thereof) and misoprostol resulted in teratogenic anomalies represented by the asymmetrical distribution of fetuses in both uterine horns, resorption sites, intensive bleeding, complete absence of fetus on one uterine horn, absence of boundaries between fetal balls, and small fetal balls. Additionally, their fetuses suffered from IUGR, morphologic malformations, hematoma, transparency of skin, and intensive bleeding. In line with our findings, a previous documentation of teratogenicity and reproductive toxicity signs induced in pregnant rats exposed to pesticides on uterus and fetus investigation [31]. As a reinforcement to our data, unsuccessful use of misoprostol as an abortifacient was associated with preterm delivery, uterine rupture, and congenital malformations [15,16].

Estrogen and progesterone are cornerstones of healthy successful gestation with normal fetal growth [32]. Both hormones increase throughout pregnancy compared to unpregnant states, followed by a sharp decline in progesterone levels towards the end of gestation for labor induction [33]. Affecting these steroidal hormone trajectories, such as their production, transport, and receptors, may result in abortion (intended) or miscarriage (unintended), and IUGR [34,35]. In the current work, treatments with MP extract and misoprostol significantly affected serum sex hormones, which may be a reason for the complete abortion or fetal resorption observed in treated rats. In agreement with a previous study that demonstrated misoprostol power to modulate serum level of estrogen and progesterone to induce abortion, and misoprostol-mediated structural abnormalities in survival fetus [14]. Furthermore, all the fractions showed an anti-progesterone effect, whereas But. and the Rem.aq. exhibited additional definite estrogenic activities. This fluctuation in sex hormone levels were manifested by the obvious growth resistance and the increased fetal resorption that was documented in their groups. The findings reported herein imply that a possible abortifacient effect of MP is to decrease progesterone levels and increase estradiol levels.

MicroRNAs (MiRs) are non-coding small molecules known as “fine-tuners” rather than “switches” as they balance gene expression in the posttranscriptional state [36,37]. Nowadays, MiRs are gaining significant attention and became a field of study by researchers targeting its implication in the disease progression and maintaining tissue homeostasis. On the other side, MiRs titers during normal pregnancy or disorder states, such as abortion, preterm labor, pre-eclampsia, IUGR, and congenital deformity did not receive the same recognition. Hence, investigating them will aid in establishing new therapeutic and diagnostic targets in pregnancy related complications. A single change in miRNA expression, at any stage of placental or fetal development, can lead to serious complications which was observed herein. All the treated rats exhibited disrupted MiRs expression that may be one of the contributing factors to IUGR, fetal resorption, and even complete abortion. The administration of Rem.aq was associated with the highest up regulation in MiR-520 among fractions that sequentially resulted in additional toxicity observed in its group. Previous studies documented that MiR-520 upregulation in both placenta and maternal circulation is highly linked to placental pathologies, such as IUGR [38] and our records of declined fetal weight supports this fact. Concerning MiR-146a, its up-regulation is highly associated with balanced micro-environment of pregnancy, and its downregulation was involved in unexplained spontaneous abortion [39] in agreement with our data.

Substances that mimic or stimulate different MiR-520 isoforms regulate and increase estradiol [40] and that is concurrent with our findings where misoprostol, MP extract, and its fractions upregulated MiR-520 that, in turn, increased estradiol (Figure 3II), resulting in abortion or IUGR. Concerning miRNAs-146, formerly it was documented that mature miRNAs-146 in endometrial epithelial cells are upregulated by progesterone that, in turn, will antagonize estrogen in endometrial epithelium [41], all these facts were reverted by MP (extract and fraction) and misoprostol administration to induce IUGR, fetal morphological deformity, or even complete abortion, where MP; total extract and or its fractions, and misoprostol decreased progesterone that resulted in MiR-146a down-regulation resulting in inhibition of the posttranscriptional regulation of estradiol and increasing its levels.

Matrix metalloproteinases (MMPs) are the most important among the classes of proteolytic enzyme in matrix degradation with similar structures and variety of functions [42]. MMPs show elevated levels during normal pregnancy, as a part of essential regulation of vascular and uterine remodeling, since significant changes in the hemodynamic and uterus is required to allow adequate uteroplacental blood flow and uterine expansion for appropriate fetal growth [43]. MMP-9 upregulation has been implicated in placentation, vasodilation, and uterine expansion during normal pregnancy. Hence, our investigation probed MMPs expression and its regulator, the tissue inhibitors of metalloproteinases (TIMPs), where MMPs are inactivated through TIMPs [43], that is produced by the same cells that produce MMP-9. In this study, all the pregnant rats treated with misoprostol, MP (250 mg/kg) extract, and fractions thereof at a dose of 125 mg/kg, demonstrated a marked inhibition in MMP-9 protein expression because of upregulation of its inhibitor the TIMP-1 protein expression. Previous studies reported the decreased MMP-9 levels may be involved in the IUGR pathogenesis during pregnancy [44], and that was evident in this work where all the fetuses showed retarded growth expressed in reduced fetal weights comparable to fetuses of untreated mothers. Moreover, MMP-9 levels are affected by pregnancy hormones (progesterone and estradiol) [45] as it was depicted formerly that progesterone regulates MMP expression/activity, which helps in management of the initiation, maintenance, and progression of endometrium [46]. Hence, the MP associated disturbance in both hormones may be the reason behind the elevated TIMP-1 and the decrease in MMP-9 uterine expression.

A certain amount of inflammatory response is paramount to normal healthy gestation, however, its exaggeration by the imbalance between pro-inflammatory (TNF-α, IL-6, and IL-8) and anti-inflammatory (IL-10, and IL-4) cytokines/chemokines, and the stimulation of different signaling molecules may sequel in many complications, such as preterm labor, IUGR, and spontaneous abortion, which may even increase the risk of morbidity and mortality to both the mother and their offspring [47,48]. Furthermore, increasing maternal inflammation through periods of fetal development is considered as a significant risk factor for birth deficits, morphological deformity, congenital diseases, and even still birth [49,50]. In our study, a marked inflammatory state in both, misoprostol and MP treated rats was manifested by elevation in the levels of TNF-α and IL-1β which participates in the observed fetal deformity in different groups, in agreement with previously published data [51]. Notably, the cytokine IL-1β was the first parameter to be linked to infection-induced premature labor. In addition, preterm labor and delivery were induced by IL-1β administration in experimental animals [52].

Phytochemical constituents in MP possess potential modulation of circulatory maternal hormones may trigger the inflammatory response as progesterone hinders the inflammatory related markers and molecules and keep the gestation environment protected [53]. Furthermore, the exaggerated inflammatory response observed herein after MP administration rationaled the disturbance in MMP-9/TIMP-1 expressions in their relevant treated groups (extract and fractions). Supporting this hypothesis, a previous study documented that overexpression of TNF-α, reduce MMP-2 and MMP-9, while an aggravation of the severity of abortion in rats further prompted TNF-α expression that sequentially decreased MMP-2 and MMP-9 levels [54]. In addition, using anti-inflammatory medication such as lactoferrin was able to prevent miscarriage/abortion by modulating MMP-9/TIMP-1 ratio and promoting repair [55]. Collectively, MP shifted the inflammatory state toward stimulation that increased risk of abortion or fetal resorption, IUGR, and morphological deformity.

Mood disorders *viz* depression and anxiety, addiction, and attempted suicide are mental health problems associated with abortion and placental pathologies induced by either stress or drugs [56]. Hence, brain investigations in this milieu have preoccupied researchers, and still there is no definite explanation available for its incidence nor prevention [57,58,59]. In this study, an open field test (OFT) was employed to examine the mental and behavioral status of the pregnant female rats under investigation. OFT is a widely used method of assessing rodent exploratory behavior, anxiety-like behaviors and determining whether a substance is sedative, poisonous, or stimulating. Consequently, the OFT assesses a variety of behavioral characteristics in addition to locomotion [60,61,62,63].

Taking into consideration all the analyzed parameters in OFT, only EtOAc and Rem.aq. treatment showed contrast behavioral effects, EtOAc treated rats showed depressive behaviors whereas Rem.aq. treated ones presented a state of anxiety and stress. Additionally, misoprostol treated group showed similar behavior to EtOAc treated group, even more pronounced. Several studies agree with the records of OFT variables and our interpretation [64,65,66]. In addition, this variance in behavior outcome, depression and/or anxiety, is common following early pregnancy loss or termination [67].

Glial fibrillary acidic protein (GFAP), is a major intermediate filament protein of differentiated astrocytes that plays a fundamental role in brain activity, not only in pregnancy but also physiologically as it modulates astrocytic motility and shape by supplying structural stability to astrocytic processes [68]. A previous study monitoring the changes in GFAP expression in the brain of rats during pregnancy and the beginning of lactation reported that GFAP expression is markedly increased on days 18 and 21 of gestation and is a sequence to changes in circulating sex steroid hormones. In the current study, administration of MP extract and its fraction on day 15th of gestation maintained GFAP cortical content on day 18th, except for Rem.aq., which showed a reduced GFAP content. Misoprostol treated rats presented the worst GFAP contents, implying a higher hazard/deleterious potential and this is the first documentation of this effect according to the author knowledge. Misoprostol and Rem.aq. changes in sex steroid hormones distinctly decreased GFAP cortical contents. Further, the decreased levels of GFAP may be in charge of the behavioral disturbance observed in rats treated with them [69].

Placental pathologies, mediated by altered MiRs levels, target gene expression of various signaling molecules that are important not only for fetal neurodevelopment but also for maintaining a normal maternal brain environment [70]. MiRs and other secreted compounds from the preeclamptic placenta showed direct effects on neurons and astrocytes [70], accordingly the changes of GFAP contents, may be related to misoprostol and Rem.aq. higher potential to modulate MicroRNAs.

Brain-derived neurotrophic factor (BDNF) is member of nerve growth factor family that has a structural role in neuronal development and function that is closely related to mood and behavioral disorders in many conditions [71]. All the treatments, except for But., decreased cortical contents of BDNF as compared to those of positive control reflecting neurogenesis inhibition. The reason behind this reduction may be the disturbance in hormonal status caused by both medical or herbal treatments for abortion induction [72,73]. Additionally, the decline in the cortical contents of BDNF contributes to the disruption in the behavior of pregnant rats as evidenced by OFT behavioral records after the different treatments. This was reinforced previously that the decline in BNDF levels have been linked to faster cognitive decline, poor memory performance, learning ability and even anxiety related behavior [62].

Serotonin, or 5-Hydroxytryptamine (5-HT) is the most manipulated monoamine for its role as a neurotransmitter in experimental animal models to establish its role in humans physiologically and pathologically. Misoprostol, MP extract and only EtOAc fraction increased cortical content of 5-HT, but the rest of the fractions did not affect brain 5-HT levels. Toxic placental effects perceived by these groups may also have a role in this monoamine increase. Our findings are supported by the theory which states that maternal mood and behavior are controlled by the placenta and that exposure to stressors or pathological conditions may result in an increase in 5-HT [74]. Additionally, altered serum progesterone and estradiol levels marked herein, participated in the increased 5-HT cortical content. As the regulation and modulation of neuronal excitability and neurotransmitter systems (5-HT is a one member of this system) is under the control of neurosteroids, steroid hormones with activity in the nervous system [75].

During the last century, the estrogenic properties of certain plants were recognized, initially in grazing animals, following the ingestion of clover revealed to contain high levels of the isoflavone compounds; formononełin and biochanin A [76]. Ever since, the term phytoestrogens have been established, referring to polyphenolic, non-steroidal plant constituents exerting estrogen-like biological action. Increasing epidemiological and chemical investigations pinpointed phytochemical classes viz. isoflavones, stilbenes, lignans, terpenes, and certain saponins as phytoestrogen candidates [77]. Phytoestrogens intake can modulate the endocrine system either by acting as agonists/antagonists for the estrogen receptors or by interfering with the estrogen biosynthesis process [78]. Apart from the estradiol-activating pathway, phytoestrogens can also exhibit their effects through non-genomic mechanisms, via interactions with cell surface receptors or by epigenetic mechanisms [41].

Like other members of the mint family, administration of *Mentha pulegium* L. extract and oil are known to pose fluctuations in estrogenic/progesterone levels and are used for terminating pregnancy [78,79]. The metabolic profile of MP extract reported herein in the results Section 2.3, encompasses a plethora of phytochemical constituents that were reported to possess effects on sex hormones.

Caffeic acid conjugates and depsides are the major water-soluble components detected in MP extract using UPLC-ESI-MS-TOF. A wide variety of pharmacological activities of these bioactive compounds have been evaluated including modulation of estrogen receptors [80,81]. Rosmarinic acid, a caffeic acid depside synthesized ubiquitously in the Lamiceae family and regarded as a marker, was reported to increase serum estradiol contents in diabetic and estrogen-deficient rats [82,83].

The metabolite profile of MP extract revealed the abundance of the flavone apigenin and several of its conjugates in addition to amentoflavone, which are reported in pennyroyal for the first time. Amentoflavone, a biflavonoid composed of two apigenin molecules, as well as its derivatives, were reported to be estrogen receptor agonists and inhibitors of aromatase enzymes in various experimental models [84,85]. A major metabolite of MP profile also was the flavone compound acacetin, which was proclaimed to be estrogen receptors agonists promoting uterine weight gain in mice [86]. Isoflavones are the most extensively studied compounds with established phytoestrogen activities [87], are represented in MP profile by a single constituent viz. daidzein-hexoside, enriched in the But. and Rem.aq. fractions. Daidzein was recorded to inhibit progesterone secretion, decrease the mRNA expression of α and β- estrogen receptors in medium ovarian follicles of pigs [88]. Prenylated flavonoids are bioactive metabolites that possess a prenyl (isoprene) group attached to the flavane nucleus identified in about 37 genera of plant families *viz*. Leguminosae, Rutaceae, Guttiferae, and Apiaceae [89]. Prenylated flavonoids are mainly regarded as phytoalexins produced in response to pathogenic microorganism offense [90] and were identified in MP profile herein for the first time owing to the high sensitivity offered by the mass detection. These biometabolites are regarded as potent phytoestrogens with 8-Prenylnaringenin capacity to compete with 17 β-estradiol for estrogen receptors binding at a dose of 10 µM [89]. Although the analytical procedure adopted herein is not enough to assign the prenylation position, it is worth mentioning that the substitution at C8 is crucial for the activity, and 6-prenylnaringenin was proven to exert a much lower estrogenic activity than the 8-prenylated isomer [91].

Quinones are another group of metabolites reported previously in Salvia miltiorrhiza [92], and detected in MP extract and its fractions (Table 3) for the first time in this study. Tanshinone IIA, a reported potent phytoestrogen with active conformation analogous to 17 β-estradiol and high binding affinity to estrogen receptors, was detected in the 70% ethanol extract of MP, MecH and Rem. aq. fractions [93].

It is worth mentioning that all MP fractions imposed fetal morphologic malformations, if not complete abortion. In brief, all the fractions presented a significant reduction in fetal weight especially with the But. and rem. Aq. fractions in addition to the highest intrauterine growth retardation (IUGR). Both fractions also caused elevation in the levels of estradiol with a decline in serum progesterone levels as compared to healthy pregnant ones and those treated with MP extract. The rem. Aq. fraction showed the highest increase in MiR-520 and TNF-α values, compared to MP, whereas MecH fraction exhibited the highest effect on TIMP-1 and MMP-9 expressions. EtOAc treated rats showed depressive behaviors contrarily to Rem. aq. treated ones which demonstrated a state of anxiety and stress. Misoprostol treated groups showed similar, or even more pronounced behavioral state to EtOAc treated group. MP extract and its fractions caused a reduced GFAP content, and misoprostol resulted in the least GFAP contents. Misoprostol, MP extract, and EtOAc fraction increased cortical content of 5-HT. All the treatments, except for the But. fraction, decreased the cortical contents of BDNF as compared to those of positive control reflecting neurogenesis inhibition.

## 4. Conclusions

The data of the current study verifies and documents the traditional reputation of employing *Mentha pulegium* L. (Pennyroyal) as an abortifacient by modulating pregnancy hormone levels, gene/protein expression, and the inflammatory process. All these cues afflicted the fetal morphology and the mothers’ brains and behavior. Indeed, MP was successful in terminating pregnancy with minimal behavioral abnormalities and low toxicity margin, yet contrarily to misoprostol, MP extract in the dose of 250 mg/kg maintained cortical contents of GFAP. UPLC-ESI-TOF-MS examination revealed MP and its fractions to encompass several metabolites with phytoestrogenic potential and mediating for the abortifacient effect. The findings reported herein encourage the future perspective of possible use of MP extract in contraception if administered at an earlier phase of the estrous cycle. Major phytochemical constituents identified in this study *viz.* rosmarinic acid, hesperidin and ursolic acid can be examined for their individual effects following the same presented model. The workflow reported in this study can be employed to investigate the safety profile of other edible or medicinal plants, particularly the herbs of family Lamiaceae, which are commonly consumed during pregnancy.

## 5. Materials and Methods

### 5.1. Plant Material

Aerial parts of *Mentha pulegium* L. (pennyroyal) were collected from El-Beheira Governorate in May, after blooming of the flowers. The taxonomic authentication was confirmed by Dr. Abdel Halem Abdel El Mogali, Professor in flora and phytotaxonomy research, Agricultural Research Centre, where a voucher specimen No. (M.P. 998) is kept. Pennyroyal was dried in the shade then pulverized, and 200 g of the powdered plant was prepared by maceration in 70% ethanol. The extract was concentrated on a rotary evaporator (Buchi R-210 evaporator, Flawil, Switzerland) until completely dry, yielding 45 g for biological investigations and UPLC-ESI-TOF-MS analyses. Portion of AE (25 g) was reconstituted in water and partitioned with increasing polarity organic solvents, as methylene chloride (MecH), ethyl acetate (EtOAc), and butanol (But). Each organic fraction was carefully evaporated until completely dry, and the remaining liquor (Rem. aq.) was also concentrated under vacuum and freeze dried for UPLC-ESI-TOF-MS examination. Dried fractions were freshly dissolved in distilled water by sonication and 2 drops of tween 80 if needed before oral administration to the rats.

### 5.2. Chemicals and Reagents

Thermo-Fisher Scientific Co. (Waltham, MA, USA) supplied the solvents acetonitrile and methanol (HPLC grade), while Sigma–Aldrich Co. (St. Louis, MO, USA) supplied the mobile phase solvents ammonium hydroxide, formic acid 98 %, and misoprostol standard drug. Analytical grade solvents (methylene chloride, ethyl acetate, and butanol) used in preparing the fractions were also purchased from Sigma-Aldrich Company.

### 5.3. Experimental Design

#### 5.3.1. Part I: Dose Selection

A dose range-finding study was conducted to screen and select the optimal abortive dose of MP extract using 48 mature albino rats, comprised of40 females (180–200 g) and 8 males (230–250 g) [94] purchased from the National Research Centre, Giza, Egypt and housed in clean plastic cages open to the room environment temperature of 24 ± 1 °C, humidity (55 ± 5%), exposed to 12 h light/dark cycle, and allowed free access to food and water throughout the study. The study was approved by Misr University for Science and Technology’s institutional ethics committee for the use of laboratory animals with approval No. PT13 on 6 March 2021 and was supervised by the Guide for the Care and Use of Laboratory Animals (NIH publication, 1996).

After acclimation for one week, mating was ordered in a 1:5 ratio, where one male rat was introduced into five caged fertile female rats overnight [95]. Mating was confirmed on the next day in the early morning by carefully inserting a cotton-tipped swap moistened with normal saline into the vaginal cavity, then withdrawing after gently rolling it against the vaginal wall [14,96]. Successful mating is marked by the presence of vaginal plugs, positive animals were considered pregnant, and this day was considered as their first day of gestation [97,98,99]. Pregnant females were also confirmed by non-invasive techniques as monitoring the physical appearance of the enlarged abdomen and eventually the noticeable body mass increase [95,100]. Hence, we used six additional un-mated female rats as a negative control to compare changes in weight through the experiment, and two of them were sacrificed at the end of the experiment, and their uteruses were preserved in formalin for histopathological examination.

On the 15th day of gestation, female rats with confirmed pregnancies were randomly divided into five groups as follows: Group I: pregnant rats serving as positive control group. Group II: rats were treated by misoprostol (100 ug/kg; once per day orally for three days) to induce abortion and served as standard control group. This dose was selected after a pilot study (data not included), guided by previous studies used misoprostol alone in abortion induction [14,101]. Group III, IV, and V received 125, 250, and 500 mg/kg of MP extract, respectively (once per day for three days starting on day 15th). The number of the completely aborted rats per group, abortion percent were calculated, the number of fetuses remaining were counted, and serum progesterone was leveled. These parameters along with the histopathological examination (*n* = 2/per group) were used for the assessment of the optimal abortion dose of different doses of MP extract.

#### 5.3.2. Part II: Identification of the Abortifacient Mechanism of MP Extract and Fractions Thereof

In this part, the sample size of male rats was increased as a conclusion from the different performed trials in the dose selection part where mating ratio 1:5 was associated with higher pregnancy failure and time consuming. Consequently, 20 male rats were randomly used for the mating of 42 female rats with a ratio 1:1 with the same previously described procedures of pregnancy confirmation in the previous section (dose selection). After the verification, the positive female rats were randomly allocated as follows: Group I: pregnant female rats that is served as positive control; Group II: rats in this group received misoprostol (100 ug/kg, orally) on day 15th of gestation; Group III: received the optimal dose of MP extract selected according to the results perceived from the first part of the study; 250 mg/kg once per day for three consecutive days starting on day 15 till day 17; Groups IV, V, VI, and VII received different fractions of MP orally (125 mg/kg; MecH, EtOAc, But., and Rem. Aq., respectively).

### 5.4. Tissue and Serum Preparation

On the 18th day of gestation, all animals’ blood samples were collected via the retro-orbital sinus under anesthesia, followed by scarification by cervical dislocation, then decapitated. The uterus, placenta, and brain cortex were rapidly removed and washed with saline. The fetuses were isolated from the uterus of different groups to screen for the changes in weight and morphology to healthy ones correspondingly.

### 5.5. Behavioral Open Field Test (OFT)

The test was commenced 24 h after the last treatment. The apparatus matches the specifications described by previous studies [102,103]. Each rat was placed individually in the center of the apparatus to be tested, the apparatus was cleaned with 70% ethanol and allowed to dry prior to each individual trial. A video camera was placed in the center of the arena above the apparatus to record the experiment for the subsequent analysis of (i) latency time (time spent from placing the animal into the center of the device till its first move), (ii) ambulation frequency (number of squares crossed by the animal over 3 min.), (iii) rearing frequency (number of times the animal stood extended and extended on rear appendages with no forelimb support), and (iv) grooming frequency (number of times of rapid cleaning movements of the forelegs towards the face and/or the body (licking the fur, washing the face, or scratching behavior) as guided by a previous study [104].

### 5.6. Parameters Assessed by ELISA Technique

As mentioned in Appendix B.

### 5.7. Histopathological Examinations

As mentioned in Appendix C.

### 5.8. Quantitative Real-Time PCR for miR-520 and miR146a Placental Expression

As mentioned in Appendix D.

### 5.9. Western Blotting

As mentioned in Appendix E.

### 5.10. Statistical Analysis

GraphPad Prism software (version 5.0d; GraphPad Software, Inc., San Diego, CA, USA) was used for all statistical analyses. The data were articulated using analysis of variance (ANOVA), followed by Tukey’s post hoc test. A Two-Way Repeated Measures (RM) ANOVA with one factor repetition (time: day) was employed to analyze the change in rat weight. The non-parametric data were expressed as the median (min–max) and were analyzed using the non-parametric Kruskal–Wallis followed by Dunn’s as the post hoc test. All the values were expressed as the mean of six rats ± S.D.A and the difference at *p <* 0.05 was considered statistically significant.

### 5.11. Metabolite Profiling of Pennyroyal (MP) Extract and Active Fractions by UPLC-ESI-TOF-MS

#### 5.11.1. Sample Preparation

Stock solutions of pennyroyal (MP) extract and the active fractions (MecH, EtOAc, But., and Rem. Aq.), were prepared by separately dissolving 50 mg of each extract in 1 ml of the solvent mixture (Water: Methanol: Acetonitrile, 50:25:25 *v*/*v*). The samples were vortexed for 2 min, ultrasonicated for 10 min, and centrifuged at 10,000 rpm for 5 min, to obtain a final concentration of 2.5 µg/µL and 10 µL of the prepared solution was injected. The LC-MS analysis procedure was also used for blank samples, quality control samples, and internal standards used for experiment setup validation. Instrument setup, spectral acquisition, and MS-data processing were adjusted as mentioned in [105,106], and provided in the Appendix A. Mass spectral data of identified metabolites in both ionization modes were compared to available online libraries and databases, *viz.* Human Metabolome Database and Pubchem. In addition, retention times and fragmentation patterns were compared to available reference standards.

#### 5.11.2. Instrument and Spectral Acquisition

As mentioned in Appendix F.

## Figures and Tables

**Figure 1 toxins-14-00347-f001:**
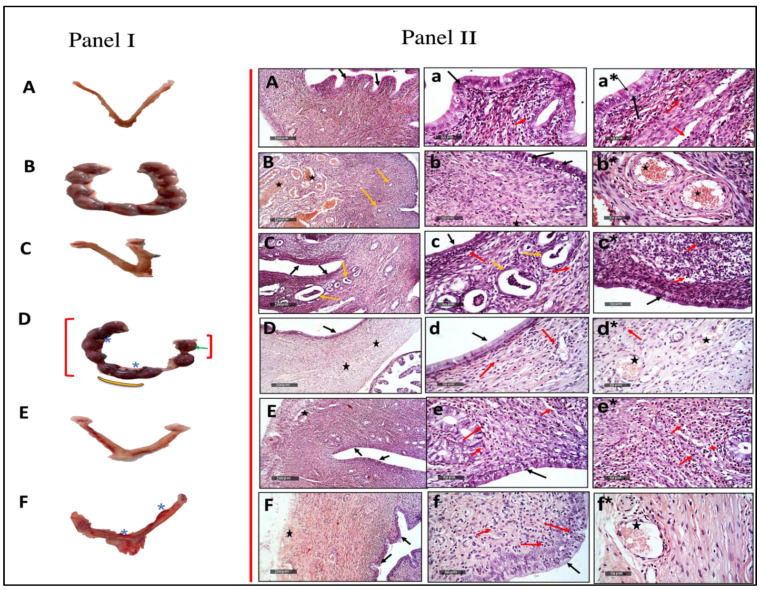
Representative images of the female rat uterus dissected on the 18th day of gestation. (**C**) misoprostol, and (**E**,**F**) MP (extract; 250, 500 mg/kg, respectively) uterus with complete abortion but (**F**) with additional appearance of intensive bleeding and hemorrhage (star) as compared to: (**A**) normal uterus presentation of unmated female rat, and (**B**) uterus with normal fetal distribution and normal physiological appearance of pregnancy. However (**D**) showed no complete abortion but asymmetric distribution of fetuses in the two uterine horns: red brackets; resorption points: green arrows; hematoma: star; and absence of boundaries between fetal balls: yellow arc (**Panel I**). Photomicrographs images of histopathological uterine tissue alterations after hematoxylin and eosin staining; (**A**,**a**,**a***) un-mated/control negative, (**B**,**b**,**b***) pregnant/control positive, (**C**,**c**,**c***) misoprostol, (**D**,**d**,**d***), (**E**,**e**,**e***), and (**F**,**f**,**f***) MP extract 125, 250, and 500 mg/kg, respectively. (**Panel II**).

**Figure 2 toxins-14-00347-f002:**
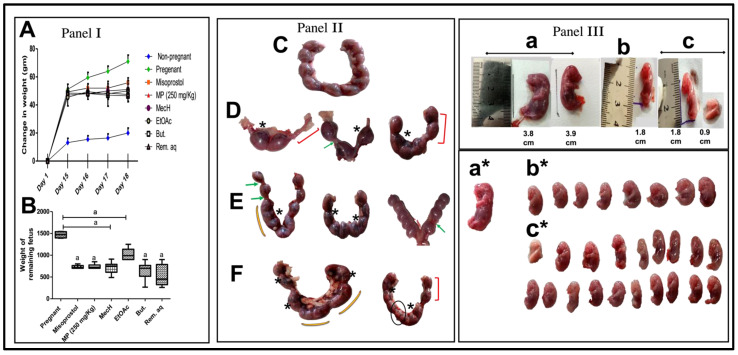
(**A**) change in weight of pregnant rats in different groups as compared to unmated/unpregnant female through the study (days: 1st, 15th, 16th, 17th, and 18th of gestation), and (**B**) weight of remaining fetus (**Panel I**). (**A**) Two-Way RM ANOVA with one factor repetition (time: day) was utilized to analyze the change in rat weight and (**B**) the non-parametric data of fetal weight were expressed as the median (min-max) and were analyzed using the non-parametric Kruskal–Wallis followed by Dunn’s as the post hoc test and student *t* test (**Panel I**). Common pathological observations and features of uterine tissues. (**C**) Pregnant rats with normal fetal distribution in the two horns; (**D**–**F**) MP (extract/fraction) or misoprostol on the 18th day of gestation. Observations: (**D**); the uterus with asymmetrical distribution of fetuses in the two horns, complete absence of fetus in one horn, remarkable resorption sites, and reduced number of fetal balls, (**E**,**F**); asymmetrical distribution of fetuses in the two horns, absence of boundaries between fetal balls, reduced sized fetal balls, notable resorption sites, and intensive bleeding. The asymmetric distribution of fetuses in the two uterine horns: red brackets; resorption points: green arrows; hematoma: star; and absence of boundaries between fetal balls: yellow arc; reduced fetal balls: black circle (**Panel II**). MP (extract or fractions) and Misoprostol-mediated structural abnormalities in fetus including height, weight, and physical appearance; (**a**,**a***) pregnant, (**b**,**b***) Misoprostol, and (**c**,**c***) MP; extract or fractions groups respectively (**Panel III**).

**Figure 3 toxins-14-00347-f003:**
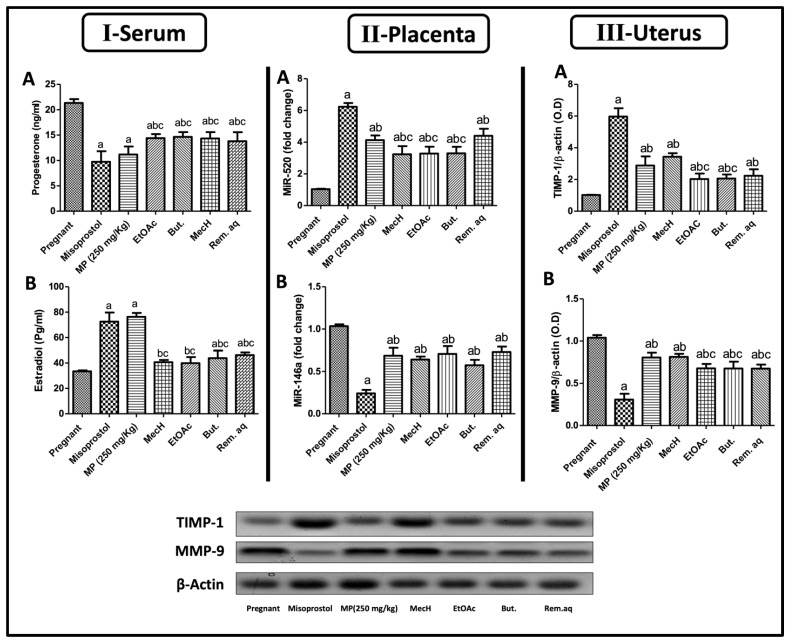
The serum levels of (**A**) progesterone and (**B**) estradiol (**Panel I**). Placental protein expression of (**A**) MiR-520 and (**B**) MiR-146a (**Panel II**). Uterine protein expression of (**A**) TIMP-1**,** (**B**) MMP-9 (**Panel III**). Values are presented as mean (*n* = 6) ± SD and statistical analysis was carried out using one-way ANOVA followed by Tukey’s post hoc multiple comparison test. As compared with control (a), misoprostol (b), MP (c); *p* < 0.05. But.: butanol; EtOAc: ethyl acetate; MecH: methylene chloride; MiR-520: micro ribonucleic acid-520; MiR-146a: micro ribonucleic acid-146a; MMP-9: matrix metalloproteinase-9; MP: *Mentha pulegium* L.; TIMP-1: tissue inhibitor matrix metalloproteinase-1; Rem. aq.: remaining aqueous.

**Figure 4 toxins-14-00347-f004:**
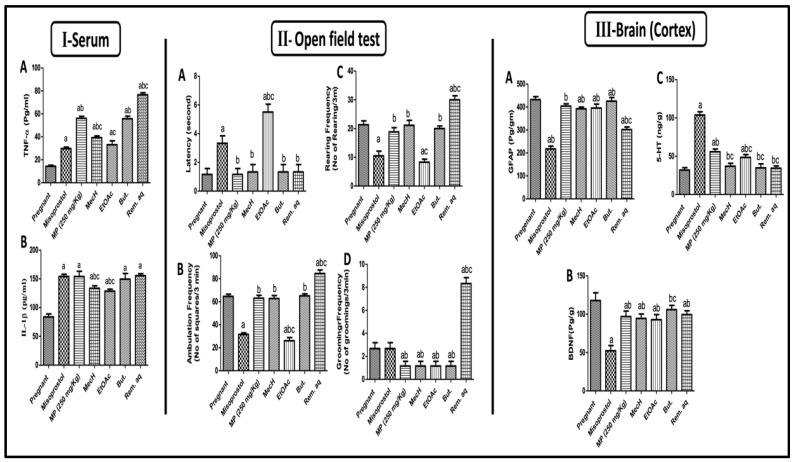
The serum levels of (**A**) TNF-α and (**B**) IL-1β (**Panel I**). The behavioral changes in an open field test on (**A**) latency time, (**B**) rearing frequency, (**C**) number of squares crossed/3 min, and (**D**) grooming frequency (**Panel II**). Cortical contents of (**A**) GFAP, (**B**) BDNF, and (**C**) 5-HT (**Panel III**). Values are presented as mean (*n* = 6) ± SD and statistical analysis was carried out using one-way ANOVA followed by Tukey’s post hoc multiple comparison test. As compared with control (a), misoprostol (b), MP (c); *p* < 0.05. BDNF: brain derived neurotrophic factor; But.: butanol; EtOAc: ethyl acetate; GFAP: Glial fibrillary acidic protein; 5-HT: 5-Hydroxytryptamine; IL-1β: interlukin-1beta; MecH: methylene chloride; MP: *Mentha pulegium* L.*;* Rem. Aq.: remaining aqueous, TNF-α: tumor necrosis factor-alfa.

**Figure 5 toxins-14-00347-f005:**
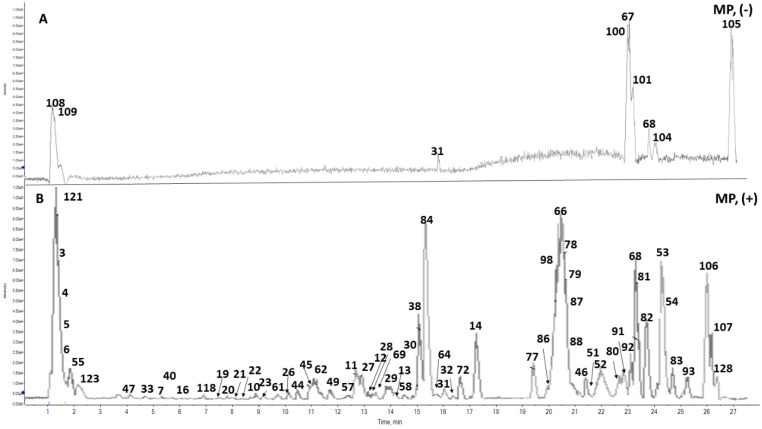
Representative UPLC-ESI-TOF-MS base peak chromatograms of 70% ethanol extract of *Mentha pulegium* L. (**A**), (−): negative ESI mode and (**B**), (+): positive ESI mode. Peak numbers follow metabolites listed in Table 3.

**Figure 6 toxins-14-00347-f006:**
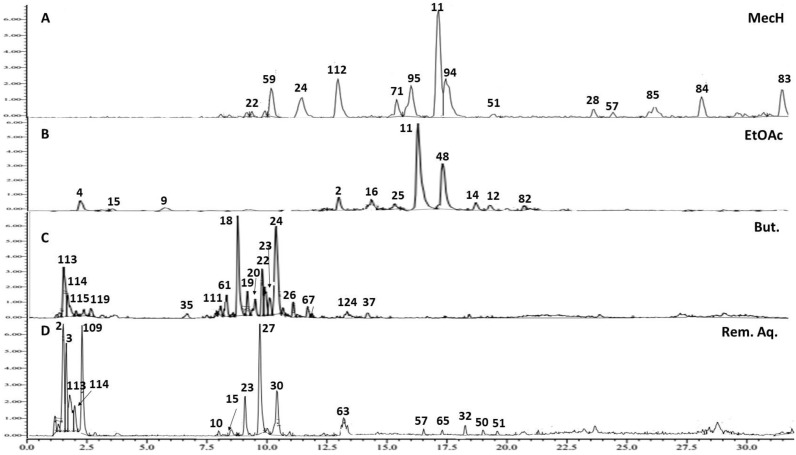
Representative UPLC/TOF-MS base peak chromatograms of fractions of *Mentha pulegium* L. (MP) in negative mode, (**A**) Methylene chloride fraction (MecH), (**B**) Ethyl acetate fraction (EtOAc), (**C**) Butanol fraction (But.) (**D**) Remaining aqueous fraction (Rem. Aq.). Peak numbers follow metabolites listed in Table 3.

**Table 1 toxins-14-00347-t001:** Effect of different doses (125, 250, and 500 mg/kg) of 70% ethanol extract of *Mentha pulegium* L. on the abortifacient parameters and serum progesterone in pregnant rats.

Group		No. of Completely Aborted Rats/Group	Abortion %	No. of Fetus Remaining	Serum Progesterone
Pregnant		0/6	0	11–12	22.5
Misoprostol 100 ug/kg		3/6	50%	7–8	9.8 ^a^
Plant Extract dose	125 mg/kg	0/6	0	8–9	18.7 ^a,b^
	250 mg/kg	2/6	33%	7–8	11.7 ^a^
	500 mg/kg	2/6	33%	7–8	10.5 ^a^

Values are presented as mean (*n* = 6) ± SD and statistical analysis was carried out using one-way ANOVA followed by Tukey’s post hoc multiple comparison test. As compared with control (a), misoprostol (b); *p <* 0.05.

**Table 2 toxins-14-00347-t002:** Effect of the 70% ethanol extract of *Mentha pulegium* L. and fractions thereof on the abortifacient parameters and serum progesterone in pregnant rats.

Group	No. of Completely Aborted Rats/gp	Abortion %	No. of Fetus Remaining
Pregnant	0/6	0	12–13
Misoprostol (ST)	2/6	33%	6–8
MP (250 mg/kg)	2/6	33%	6–9
MecH (125 mg/kg)	0/6	0	7–11
EtOAc (125 mg/kg)	0/6	0	7–11
But. (125 mg/kg)	0/6	0	3–8
Rem. aq. (125 mg/kg)	0/6	0	2–6

**Table 3 toxins-14-00347-t003:** Metabolites tentatively identified in the 70% ethanol extract of (*Mentha pulegium* L.) and its fractions methylene chloride (MecH), ethyl acetate (EtOAc), butanol (But) and remaining liquor (Rem. aq.) via UPLC-ESI-TOF-MS in both; negative/positive ESI ionization modes.

Peak #	RT. (min.)	Metabolite Name	Mol. Ion *m*/*z*		Mass (ppm)	Elemental Composition	MS^2^ Ions *m*/*z* (−)/(+)	Fractions
			[M − H]^−^	[M + H]^+^				
Hydroxycinnamic acids and derivatives								
1	1.31	Caffeic acid-*O*-glucuronide	355.0852		−5.2	C_15_H_16_O_10_	193, 161	
2	1.33	Caffeic acid	179.0561		0.3	C_9_H_8_O_4_	161	EtOAc, Rem. aq.
3	1.49	Ferulic acid		195.0856	0.8	C_10_H_10_O_4_	177, 109	MecH, Rem. aq
4	1.58	Caffeic acid dimer	341.1071	343.0823	−2.2/3.2	C_18_H_14_O_7_	179, 161/181, 163	EtOAc
5	1.87	Sinapoyl-*O*-hexoside		387.1408	1.9	C_17_H_22_O_10_	255, 169	Rem. aq
6	1.89	Cinnamic acid		149.0949	4.1	C_9_H_8_O_2_	121, 65	
7	5.17	Ethyl caffeate		209.0811	1.4	C_11_H_12_O_4_	191, 166	
8	7.26	Salvianolic acid K		557.1252	−6	C_27_H_24_O_13_	363, 345	But.
9	8.82	Caffeoyl-*O*-sinapoylquinic acid	559.1422		−4.3	C_27_H_28_O_13_	490, 354, 287	MecH, EtOAc
10	8.98	Salvianolic acid F		315.0836	−7.8	C_17_H_14_O_6_	209, 179, 167	Rem. aq
11	12.55	Rosmarinic acid		361.0907	−2.9	C_18_H_16_O_8_	331, 313	MecH, EtOAc
12	13.30	Caftaric acid		313.2372	7.8	C_13_H_12_O_9_	295, 272, 259, 137	EtOAc
13	14.42	Methyl rosmarinate		375.1082	1.9	C_19_H_18_O_8_	360, 345, 197	EtOAc
14	18.03	Rosmarinic acid derivative		379.2811	−2	C_23_H_38_O_4_	361, 319, 165	EtOAc
Flavones and derivatives								
15	4.87	Apigenin-6,8-di-*C*-hexoside	593.1489		3.5	C_27_H_30_O_15_	473, 353	Rem. aq, EtOAc
16	5.18	Penduletin-4′-*O*- glucuronide		521.1812	−1.5	C_24_H_24_O_13_	503, 345, 327, 253	EtOAc
17	7.15	5-Hydroxy-6,7,4′-trimethoxyflavone (Salvigenin)	327.2178		3.6	C_18_H_16_O_6_	190, 171	MecH
18	7.36	Apigenin-7-*O*-rutinoside (Isorhoifolin)	577.1605		9.1	C_27_H_30_O_14_	464, 269	But.
19	7.67	Unknown apigenin glycoside		563.1752	−1.3	C_27_H_30_O_13_	401, 383, 271	But.
20	7.79	Diosmetin-7-*O*-rutinoside (Diosmin)	607.1656	609.1782	−2.4/−0.3	C_28_H_32_O_15_	299/463, 301	But.
21	7.85	Luteolin-7-*O*-glucuronide		463.0899	6.1	C_21_H_18_O_12_	446, 287	
22	8.52	Apigenin-7-*O*-glucuronide	445.0776	447.0827	−7.3/1.1	C_21_H_18_O_11_	269, 175/271	MecH, But.
23	9.35	Acacetin-7-*O*-rutinoside (Linarin)		593.1876	1.9	C_28_H_32_O_14_	447, 285	But., Rem. aq
24	10.11	5,6,7-Trihydroxyflavone (Baicalein)	269.0448		1.1	C_15_H_10_O_5_		But., MecH
25	10.32	Pedalitin tetraacetate	483.0909		−2.6	C_24_H_20_O_11_	336, 309	EtOAc
26	10.42	5,7,3′-trihydroxy-4′-methoxyflavone (Diosmetin, 4′-Methylluteolin)	299.0927	301.1382	−1.5/2.1	C_16_H_12_O_6_	284, 151/283	But.
27	13.11	5,7-Dihydroxy-4′,6-dimethoxyflavone (Pectolinarigenin)		315.0857	−2.1	C_17_H_14_O_6_	300, 282, 254	Rem. aq
28	14.02	5,4′-Dihydroxy-3,6,7-trimethoxyflavone (Penduletin)		345.0946	−6.6	C_18_H_16_O_7_	330, 315, 197	MecH
29	14.23	5,7-Dihydroxy-4′-methoxyflavone (Acacetin)	283.0611	285.0753	3.5/−1.5	C_16_H_12_O_5_	268, 151/242, 153	MecH
30	15.33	5-hydroxy-3,7,3′,4′-tetramethoxyflavone (Retusin)		359.1128	0.8	C_19_H_18_O_7_	326, 162	EtOAc, Rem. aq
31	16.41	5-Hydroxy-3,3′,4′,6,7-pentamethoxyflavone (Artemetin)	387.1095	389.1230	5.2/−0.3	C_20_H_20_O_8_	340, 319/359, 341	EtOAc, But., Rem. aq
32	16.88	5-hydroxy-7,4′-dimethoxy-6,8-dimethylflavone (Eucalyptin)		327.1252	8.0	C_19_H_18_O_5_	277, 137	But., Rem. aq
Flavanone and Flavanol derivatives								
33	5.07	Naringenin-7-*O*-glucuronide		449.1112	7.6	C_21_H_20_O_11_	357, 273, 181	
34	5.37	Isosakuranetin-*O*-rutinoside (Didymin)		595.2831	3.6	C_28_H_34_O_14_	577, 457	
35	5.93	Hesperetin	301.2005		−1.5	C_16_H_14_O_6_	283, 255	But.
36	8.16	Eriodictyol-7-*O*-glucuronide	463.0990		3.2	C_21_H_20_O_12_	354, 286, 218	But.
37	13.09	Sakuranetin	285.0781		8	C_16_H_14_O_5_	267, 164	But.
38	15.45	Taxifolin		305.0845	8.3	C_15_H_12_O_7_		
39	21.51	Hesperetin-7-*O*-rutinoside (hesperidin)		611.2864	−7.5	C_28_H_34_O_15_	567, 538	
Prenyl flavones								
40	5.66	Prenyl pinocembrin		325.1398	1.4	C_20_H_20_O_4_	307, 191	
41	7.24	Prenyl kaempferol	353.2329		1.9	C_20_H_18_O_6_	285	
42	7.93	7-O-methyl isoxanthohumol		369.1311	−5.9	C_22_H_24_O_5_		
43	8.21	7,4′-Di-*O*-methyl isoxanthohumol		383.184	−3.4	C_23_H_26_O_5_	365, 233	
44	10.88	Isoxanthohumol		355.1521	−5.3	C_21_H_22_O_5_	267, 163	
45	14.99	Prenyl naringenin	339.216	341.1379	6.5/−1.3	C_20_H_20_O_5_	309, 265	
46	21.76	Dorsmanin F		441.1897	−2.5	C_25_H_28_O_7_		
Isoflavone								
47	4.35	Daidzein-8-*C*-hexoside		417.1302	−4.6	C_21_H_20_O_9_	267, 255	But., Rem. aq
Flavonols								
48	7.35	Myricetin-3-*O*-glucuronide		495.2094	4.1	C_21_H_18_O_14_	319, 301, 283	EtOAc
49	11.44	3,5,7,3′,4′,5′-Hexahydroxyflavone (Myricetin)	317.0562	319.0758	−2.7/0.6	C_15_H_10_O_8_	225, 164/151	
Biflavonoid								
50	18.88	Di-*O*-methylamentoflavone (Ginkgetin)	565.1129	567.1288	0.3/0.3	C_32_H_22_O_10_	297, 283, 165	Rem. aq
51	21.59	Amentoflavone-7,4′,4‴-trimethyl ether (Sciadopitysin)	579.1275	581.1424	−1.8/−3.2	C_33_H_24_O_10_	297	MecH
52	22.62	4′-Monomethylamentoflavone (Bilobetin)		553.2685	−3.6	C_31_H_20_O_10_	335, 473	Rem. aq
53	24.06	Unknown biflavonoid		647.2296	3.2	C_39_H_34_O_9_	629	
54	24.98	Isochamaejasmin		543.1318	4.9	C_30_H_22_O_10_	381, 322, 122	
Quinones								
55	11.49	Przewaquinone C		297.1134	4.1	C_18_H_16_O_4_	279, 261	Rem. aq
56	12.76	Przewaquinone A		311.1283	1.7	C_19_H_18_O_4_		EtOAc, Rem. aq
57	12.78	Tanshinone IIA		295.1339	3.5	C_19_H_18_O_3_	277, 149	MecH, Rem. aq
58	15.07	Przewaquinone F		313.1075	1.5	C_18_H_16_O_5_	277, 259, 149	Rem. aq
Iridoids								
59	5.54	Loganic acid		377.1466	6.4	C_16_H_24_O_10_		MecH
60	5.66	Aucubin		347.1309	−6.6	C_15_H_22_O_9_	329, 193	
61	9.09	Nepetalactone (Epi-nepetalactone)	165.0916	167.1074	3.8/4.5	C_10_H_14_O_2_	147, 107/149, 121	But.
62	10.61	Dihydronepetalactone		169.1218	−2.9	C_10_H_16_O_2_	151, 123, 83	
63	14.97	Kanokoside A	475.1817		1.6	C_21_H_32_O_12_	339, 271	Rem. aq
64	15.80	Loganin		391.1634	8.9	C_17_H_26_O_10_		
65	17.77	Patriscabroside I	361.1489		−1.1	C_16_H_26_O_9_	196, 165	But., Rem. aq
66	21.56	Kanokoside C		639.2484	−1.7	C_27_H_42_O_17_	621, 579, 562	
67	22.64	Deoxyloganic acid tetraacetate	527.1747		−2.3	C_24_H_32_O_13_	459, 391, 323	But.
68	23.67	Kanokoside D		625.2671	−4.9	C_27_H_44_O_16_	607, 581, 521	
Mono and diterpenes								
69	13.33	Carnosic acid		333.2042	−5.4	C_20_H_28_O_4_	315, 297	
70	13.53	Abienol		291.0687	−0.2	C_20_H_34_O	273, 217	
71	13.88	Carnosol		331.1913	2.7	C_20_H_26_O_4_	278, 203	Rem. aq, MecH
72	17.08	Ethyl abietic acid		331.1690	−0.8	C_23_H_22_O_2_	183	
73	18.28	12-*O*-Methylcarnosic acid	345.2070		2.9	C_21_H_30_O_4_	299, 277	
74	18.93	Taxodione		315.1958	0.9	C_20_H_26_O_3_	177, 123	
75	18.94	Coleonol (Forskolin)	409.2562		−5.5	C_22_H_34_O_7_	351, 341	
76	19.01	Pachyphyllone	315.1958	317.2122	3.4	C_20_H_28_O_3_	149	
77	19.60	Picrocrocin		331.1741	−3.2	C_16_H_26_O_7_	183, 149	
78	21.03	Neoandrographolide		481.2781	−3.1	C_26_H_40_O_8_	441, 401	
79	21.61	Abietic (Sylvic) acid		303.2223	−3.2	C_20_H_30_O_2_	165	
80	22.85	Casearborin E		597.2730	2.8	C_33_H_40_O_10_	579, 553	
81	23.82	Casearborin C/D		555.2575	−2.4	C_31_H_38_O_9_	527	
82	24.25	Casearborin A		539.2663	4.4	C_31_H_38_O_8_		EtOAc, But., Rem. aq
83	24.91	Dihydrotanshinone I		279.1027	4.2	C_18_H_14_O_3_	149	MecH
Triterpenes								
84	16.05	Corosolic acid		473.2323	0.1	C_30_H_48_O_4_	455	MecH
85	17.22	Asiatic acid		489.3607	6.7	C_30_H_48_O_5_	453, 407, 201	MecH, EtOAc
86	19.21	Ursolic acid methyl ester		471.3478	1.8	C_31_H_50_O_3_	425, 407	
87	20.87	Melilotoside A		591.4307	8.2	C_35_H_58_O_7_	574, 292, 133	
88	21.31	Platanic acid		459.3481	2.7	C_29_H_46_O_4_	442, 316	But.
89	21.92	Orthosiphol D		553.2685	−0.7	C_31_H_36_O_9_	525	
90	23.53	Unknown triterpene (Swietmanin I)		567.2609	3.6	C_32_H_38_O_9_		
91	22.71	Micromeric acid		455.3526	1.3	C_30_H_46_O_3_	437, 247, 203	
92	23.25	Ursolic acid		457.3684	1.6	C_30_H_48_O_3_	439, 411, 393	
93	25.26	Conrauidienol		467.3876	−1.7	C_32_H_50_O_2_	450	
Fatty acids and esters								
94	7.83	Pinellic acid	329.2337		4.2	C_18_H_34_O_5_	211, 171	MecH
95	14.89	16-Hydroxyhexadecanoic acid (Juniperic acid)	271.2267		−0.3	C_16_H_32_O_3_	225	MecH
96	17.56	Myrestic acid	227.2007		0.6	C_14_H_28_O_2_		Rem. aq
97	19.19	Palmitoleic acid	253.2173		4.3	C_16_H_30_O_2_		
98	19.31	Hydroxyoctadecatrienoic acid		295.2268	−6.7	C_18_H_30_O_3_	277, 179	
99	19.58	Linolenic acid	277.2182		−0.3	C_18_H_30_O_2_		
100	22.73	Palmitic acid	255.2318		−0.1	C_16_H_32_O_2_	237	
101	23.18	Tetracosanoic (Lignoceric) acid	367.3574		1.0	C_24_H_48_O_2_		
102	23.46	Methyl 12,13-epoxystearate		313.2753	5.0	C_19_H_36_O_3_	257, 239, 97	MecH, But.
103	23.52	Glyceryl palmitate		331.2858	4.7	C_19_H_38_O_4_	239	
104	23.77	Oleic acid	281.2485		3.6	C_18_H_34_O_2_		
105	25.55	Methyl oleate	295.2628		−1.1	C_19_H_36_O_2_		
106	26.48	3-Hydroxypropyl oleate		341.3038	−3.6	C_21_H_40_O_3_	95	
107	26.66	Eicosadienoic acid		309.2762	−8.2	C_20_H_36_O_2_	291, 109	
Aliphatic and Hydroxybenzoic acid derivatives								
108	1.18	Tartaric acid	149.0098		−2.0	C_4_H_6_O_6_		
109	1.21	Quinic acid	191.0549		−0.5	C_7_H_12_O_6_	111	Rem. aq
110	1.35	3,4-Dihydroxyphenylacetic acid	167.0007		−0.9	C_8_H_8_O_4_	148, 78	
111	1.37	2-Isopropylmalic acid	175.0588		8.6	C_7_H_12_O_5_		But.
112	1.38	Galacturonic acid	193.0709		2.0	C_6_H_10_O_7_		MecH
113	1.39	Malic acid	133.0500		−0.9	C_4_H_6_O_5_		Rem. aq, But.
114	1.40	Protocatechuic acid hexoside	315.0716		1.8	C_13_H_16_O_9_	195, 153, 109	But., Rem. aq
115	1.41	Hydroquinone glucuronide	285.0592		−4.5	C_12_H_14_O_8_	165, 152	But.
116	1.48	Hydroxyphenyllactic acid	181.0486		−5.0	C_9_H_9_O_4_	166, 112	
117	4.57	Suberic acid	173.1190		0.1	C_8_H_14_O_4_		
118	6.39	Tuberonic acid (12-hydroxy-7-isojasmonic acid)		227.1286	3.6	C_12_H_18_O_4_	209, 191, 131	
119	6.61	Pinonic acid	183.1021		2.7	C_10_H_16_O_3_	137	But.
120	16.93	Menthyl salicylate		277.1786	−4.3	C_17_H_24_O_3_	231, 137	
Others								
121	1.26	Valine		118.0862	−3.9	C_5_H_11_NO_2_	58	
122	1.68	Niacin (nicotinic acid)		124.0393	1.0	C_6_H_5_NO_2_	106, 80	
123	2.12	Proline		116.0706	−7.2	C_5_H_9_NO_2_	84, 70	
124	4.25	Oleacein		321.1337	1.2	C_17_H_20_O_6_	149	MecH, EtOAc, But.
125	4.34	Hydroxyquinoline		146.0600	−1.1	C_9_H_7_NO	118, 91	
126	8.05	Loliolide		197.1166	−3.3	C_11_H_16_O_3_	179, 105	
127	10.58	Tryptophol		162.0913	5.8	C_10_H_1_1NO	146, 118	
128	26.89	7-hydroxy-4-methyl-coumarin (Hymecromone)		177.0543	−1.6	C_10_H_8_O_3_	159, 149	But.

## Data Availability

Data are available upon request from the corresponding author (daliarasheed@o6u.edu.eg).

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
