# Peer review of "Mentha pulegium L. (Pennyroyal, Lamiaceae) Extracts Impose Abortion or Fetal-Mediated Toxicity in Pregnant Rats; Evidenced by the Modulation of Pregnancy Hormones, MiR-520, MiR-146a, TIMP-1 and MMP-9 Protein Expressions, Inflammatory State, Certain Related Signaling Pathways, and Metabolite Profiling via UPLC-ESI-TOF-MS"

_toxins, 2022, doi:10.3390/toxins14050347_

Round 1

Reviewer 1 Report

The research article titled Mentha pulegium L. (pennyroyal, Lamiaceae) extracts impose abortion or fetal-mediated toxicity in pregnant rats; evidenced by the modulation of pregnancy hormones, MiR-520, MiR-146a, TIMP-1 and MMP-9 protein expressions, inflammatory state, certain related signaling pathways, and metabolite profiling via UPLC-ESI-TOF-MS is well supported by the scientific evidence. In my opinion it is suitable for publication after incorporating the following changes.

  1. Write scientific name of the plant in italic. Please change throughout the manuscript.
  2. Always use complete name at first and then abbreviate.
  3. Explain why it is important to analyze crude extract and then subsequent fraction? Just to have better identification?
  4. Why ethanol is used for extraction as it is toxic and why water is not used for extraction purposes? Methyl chloride, ethyl acetate, and butanol are used for sequential extraction and they also possess toxicity. It is possible a part of toxicity that resulted in abortion is attributed by extraction solvents toxicity?
  5. As a result of biochemical profiling, 128 compounds belonging to several classes are being identified. Please identify and discuss more about the compounds that are toxic in your opinion.
  6. The quality and presentation of the figures are very low. Most of the words are not readable.

Abstract

Please give a background of plant use and need of project, methodology employed, results, and conclusion.

Introduction

Line 30-31. Please give reference

Line 32-34. This seems to be the study gap and move it before the aim of study at the end of the introduction

Line 65. Word controversy is not suitable here. According to cited literature, it is abortifacient but the mechanism is unknown?

Add data about the plant including its biological activities.

Material and methods

Please add the model and manufacturer of chemicals and equipment used

Write everywhere 70% ethanol extract instead of alcoholic extract  

Line 640-642. Please re-write the sentence

Line 705, Then is used twice. Please change

Results

Line 75, Italic the plant name

Line 75, Dose selection was body weight-related? Please check and add

Table 1, Please remove ST and add the dosage

Table 1, write plant extract and dosage in the second line

Line 88, write p in italic

Line 99. Change plant name to italic

Line 123, Delete results

Figure 2. Panel 1 cannot be read. Please use black letters only.

Line 166. Misoprostol concentration used?

Line 202, Please write full name at first; TNF-α and (B) IL-1β

Line 261, What is O/C?

Line 268, write are explained in the following section instead of will be explained

Table 3. Row 33 the type of extract/fraction is not mentioned. Is it butanol? If yes, then write again as a new class of compounds commences later. Please correct this throughout table 3.

It is mentioned in the material and methods that retention times and fragmentation patterns were compared to available reference standards. Please mention those compounds identified using external standards? And also those identified as using online mass banks.

In my opinion, most of the compounds are common and can be compared in a much better way using previous literature. 

Discussion

Line 487-488. Please mention the names of pro- and anti-inflammatory cytokines.

  1. Please discuss these papers wherein toxicity of said plant is described to compare your results
  • Abdelli, M., Moghrani, H., Aboun, A., & Maachi, R. (2016). Algerian Mentha pulegium L. leaves essential oil: Chemical composition, antimicrobial, insecticidal and antioxidant activities. Industrial Crops and Products94, 197-205.
  • Messaoudi, M., Begaa, S., Benarfa, A., Ouakouak, H., Benchikha, N., & Ferhat, M. A. (2020). Radiochemical separation by liquid-liquid extraction for the determination of selenium in Mentha pulegium L.: Toxicity monitoring and health study. Applied Radiation and Isotopes159, 109099.

Conclusion

Line 622-624. Re-write the sentence.

Please write few lines about future studies specifically about studied plant

Author Response

Rebuttal letter

The authors are thankful to the reviewer’s effort to improve the manuscript and here is the response to the reviewer’s comments

  • Write the scientific name of the plant in italic. Please change throughout the manuscript.

Response: Done

  • Always use the complete name at first and then abbreviate it.

Response: Checked, done and an abbreviation list (line 840) was added at the end of manuscript for convenience of the readers.

  • Explain why it is important to analyze crude extract and then subsequent fraction? Just to have better identification?

Response: The crude extract was analysed to investigate its impact on pregnant rats and establish its abortifacient effect. The fractions thereof were further examined in an endeavor to unravel if this effect was confined to any of the four tested fractions (methylene chloride, ethyl acetate, butanol or the remaining liquor), hence it would guide the authors to specify the class of phytoconstituents responsible for the mechanism of action.

  • Why ethanol was used for extraction as it is toxic and why water is not used for extraction purposes? Methyl chloride, ethyl acetate, and butanol are used for sequential extraction and they also possess toxicity. It is possible a part of toxicity that resulted in abortion is attributed by extraction solvents toxicity?

Response: The aim of this study was to investigate the secondary metabolite composition of Mentha pulegium holistically and in an untargeted manner. Choosing the aqueous extraction would include mainly the polar constituents present in the extract of pennyroyal viz. quinones, flavonoid and cinnamic acid conjugates. However, pennyroyal and other members of Lamiaceae are rich in terpene derivatives as 36 metabolites were detected herein (table 3) and they had to be included in the study by extracting the plant material using 70% ethanol. As for the solvents used in extraction, each fraction was carefully evaporated under reduced pressure till free of the solvent and complete dryness (mentioned in lines 669-670), and they were reconstituted in water before administration to the rats.  This note was not mentioned and has been added to the manuscript in lines (675-677) and the authors are grateful to the reviewer for his comment.

  • As a result of biochemical profiling, 128 compounds belonging to several classes are being identified. Please identify and discuss more about the compounds that are toxic in your opinion.

Response: The study revealed that the crude extract as well as its fractions presented morphologic malformations, hormonal modulatory action as well as complete abortion in some cases (a new paragraph was added in the discussion section lines 631-645) . These findings did not confine the abortifacient effect to a specific fraction, hence it can be concluded that the action is due to a synergistic action mediated by the various secondary metabolites of pennyroyal rather than a single compound or one specific fraction.

The new paragraph states:

It is worth mentioning that all MP fractions imposed fetal morphologic malformations if not complete abortion. In brief, all the fractions presented a significant reduction in fetal weight was detected especially with the But. and rem. Aq. fractions in addition to the highest intrauterine growth retardation (IUGR). Both fractions also caused elevation in the levels of estradiol with a decline in serum progesterone levels as compared to healthy pregnant ones and those treated with MP extract. The rem. Aq. fraction showed the highest increase in MiR-520 and TNF-α values, compared to MP, whereas MecH fraction exhibited the highest effect on TIMP-1 and MMP-9 expressions. EtOAc treated rats showed depressive behaviors contrarily to Rem. aq. treated ones which presented a state of anxiety and stress. Misoprostol treated groups showed similar, or even more pronounced behavioral state, to EtOAc treated group. MP extract and its fractions showed a reduced GFAP content, and misoprostol presented the least GFAP contents. Misoprostol, MP extract, and EtOAc fraction increased cortical content of 5-HT. All the treatments, except for the But. fraction, decreased the cortical contents of BDNF as compared to those of positive control reflecting neurogenesis inhibition.

  • The quality and presentation of the figures are very low. Most of the words are not readable.

Response: The figures have been updated with higher resolution.

  • Abstract

Please give a background of plant use and need of project, methodology employed, results, and conclusion

Response: According to the author’s guideline of the journal, the abstract should be 200 words only. The authors are ready with a more detailed abstract, but will it be accepted by the journal editorial office?

  • Introduction

Line 30-31. Please give reference

Response: Reference has been added.

  • Line 32-34. This seems to be the study gap and move it before the aim of study at the end of the introduction

Response: The section was reallocated according to the reviewer’s comment.

  • Line 65. Word controversy is not suitable here. According to cited literature, it is abortifacient, but the mechanism is unknown?

Response: Word controversy has been omitted.

  • Add data about the plant including its biological activities.

Response: Lines 38-54 include the required data.

Material and methods

  • Please add the model and manufacturer of chemicals and equipment used

Response: Done and highlighted throughout the manuscript.

  • Write everywhere 70% ethanol extract instead of alcoholic extract  

Response: Done and highlighted throughout the manuscript.

  • Line 640-642. Please re-write the sentence

Response: Lines have been rewritten and highlighted in the manuscript.

  • Line 705, Then is used twice. Please change

Response: Changed to followed by.

Results

  • Line 75, Italic the plant name

Response: DONE

  • Line 75, Dose selection was body weight-related? Please check and add

Response: The authors thank the reviewer for his remark, and the body weight was added in the manuscript in dose selection part line 75

  • Table 1, Please remove ST and add the dosage

Response: DONE

  • Table 1, write plant extract and dosage in the second line

Response: DONE

  • Line 88, write pin italic

Response: DONE

  • Line 99. Change plant name to italic

Response: DONE

  • Line 123, Delete results

Response: DONE

  • Figure 2. Panel 1 cannot be read. Please use black letters only.

Response: DONE

  • Line 166. Misoprostol concentration used?

Response: misoprostol dose (100 ug/kg) was added and highlighted.

  • Line 202, Please write full name at first; TNF-α and (B) IL-1β

Response: DONE and an abbreviation list (line 840) was added at the end of manuscript for convenience of the readers.

  • Line 261, What is O/C?

Response: refers to the nature of the glycosidic linkage; either an O-glycosidic bond, or C-glycosyl bond between the aglycone and the sugar moiety.

  • Line 268, write are explained in the following section instead of will be explained

Response: done

  • Table 3. Row 33 the type of extract/fraction is not mentioned. Is it butanol? If yes, then write again as a new class of compounds commences later. Please correct this throughout table 3.

Response: The fractions listed in table 3 were confined to the compounds having both the molecular ion peak and fragmentation pattern losses matching those observed in the total ethanol extract. Other compounds lacking the complete fragmentation were not included in the table.

  • It is mentioned in the material and methods that retention times and fragmentation patterns were compared to available reference standards. Please mention those compounds identified using external standards? And also those identified as using online mass banks.

Response: The available references included mainly the common available secondary metabolites, viz. caffeic, cinnamic and rosmarinic acids, in addition to the common fatty acids, viz. oleic, linoelic and palmitic acids. The identification is usually based on interpretation of fragmentation pattern loses and comparison to mass banks and reported literature.

  • In my opinion, most of the compounds are common and can be compared in a much better way using previous literature. 

Response: the compounds annotation was done based on interpretation of the fragmentation losses and comparison to reported literature which have been mentioned in the manuscript, references number 17-29 lines 288-401, as suggested by reviewer indeed.

Discussion

  • Line 487-488. Please mention the names of pro- and anti-inflammatory cytokines.

Response: names have been mentioned and highlighted in the manuscript. (A certain extent of inflammatory response is paramount to normal healthy gestation, however, its exaggeration by the imbalance between pro-inflammatory (TNF-α, IL-6, and IL-8) and anti-inflammatory (IL-10, and IL-4) cytokines/chemokines)

  • Please discuss these papers wherein the toxicity of said plant is described to compare your results

  • Abdelli, M., Moghrani, H., Aboun, A., & Maachi, R. (2016). Algerian Mentha pulegium L. leaves essential oil: Chemical composition, antimicrobial, insecticidal and antioxidant activities. Industrial Crops and Products94, 197-205.
  • Messaoudi, M., Begaa, S., Benarfa, A., Ouakouak, H., Benchikha, N., & Ferhat, M. A. (2020). Radiochemical separation by liquid-liquid extraction for the determination of selenium in Mentha pulegium L.: Toxicity monitoring and health study. Applied Radiation and Isotopes159, 109099.

Response: The authors are thankful to the reviewer, but the first reference discusses the effect of Mentha pulegium L. essential oil, which is known to be toxic due to richness in pulegone as mentioned in the introduction line 42 (the reference was additionally added in the manuscript for confirmation). The second reference also is beyond the scope of this study and no toxic components were reported in both studies by name.

Conclusion

  • Line 622-624. Re-write the sentence.

Response: lines have been corrected and authors are thankful to the reviewer.

  • Please write few lines about future studies specifically about studied plant

Response: done and highlighted in the conclusion (The findings reported herein encourage the future perspective of possible use of Mentha pulegium extract in contraception if administered at an earlier phase of the estrous cycle. Major phytochemical constituents identified in this study viz. rosmarinic acid, hesperidin and ursolic acid can be examined for their individual effects following the same presented model. The workflow reported in this study can be employed to investigate the safety profile of other edible or medicinal plants, particularly the herbs of family Lamiaceae, which are commonly consumed during pregnancy)

Reviewer 2 Report

The MS entitled "Mentha pulegium L. (pennyroyal, Lamiaceae) extracts impose abortion or fetal-mediated toxicity in pregnant rats; evidenced by the modulation of pregnancy hormones, MiR-520, MiR-146a, TIMP-1 and MMP-9 protein expressions, inflammatory state, certain related signaling pathways, and metabolite profiling via UPLC-ESI-TOF-MS" was thoroughly reviewed and report generated. The MS is very well composed and drafted. Amongst some of the format related suggestions:

  1. Correct the font size of “of’” in the title.
  2. The scientific names of all plants/organisms should be italic throughout the MS, including the name Mentha pulegium (e.g page 6 line 182).
  3. The figure 2 part (a) and (b) should be rearranged for clear depiction to the readers.
  4. page 5 lines 165. The section 2.2.2 should be shifted to next page.
  5. Line 336, methanolic or ethanolic? Please correct.
  6. correction in line 624 in conclusion section.
  7. Material method line 656, correct the heading format.

Author Response

Rebuttal Letter

The authors appreciate the reviewer’s efforts in revising the manuscript entitled: “Mentha pulegium L. (pennyroyal, Lamiaceae) extracts impose abortion or fetal-mediated toxicity in pregnant rats; evidenced by the modulation of pregnancy hormones, MiR-520, MiR-146a, TIMP-1 and MMP-9 protein expressions, inflammatory state, certain related signaling pathways, and metabolite profiling via UPLC-ESI-TOF-MS”. Please find below our responses to the reviewer’s comments.

Reviewer #2

Comment

Response

1.      Correct the font size of “of’” in the title.

Done

2.      The scientific names of all plants/organisms should be italic throughout the MS, including the name Mentha pulegium (e.g page 6 line 182).

Done

3.      The figure 2 part (a) and (b) should be rearranged for clear depiction to the readers.

If the reviewer means panel III:

The figure  follows the workflow, and the comparative means where ( a) is the control and (b) is the slandered and better be presented in this manner for the comparative means. but if the reviewer means something else, please inform us to display the results in the best form.

4.      page 5 lines 165. The section 2.2.2 should be shifted to next page.

Done

5.      Line 336, methanolic or ethanolic? Please correct.

Done

6.      correction in line 624 in conclusion section.

Done

7.      Material method line 656, correct the heading format

Done

Reviewer 3 Report

The Authors performed detailed studies of the potential impact of Mentha pulegium L. as potential abortion or fetal-mediated toxicity in pregnant rats factor. The topic of the manuscript is suitable for the JOurnal. Generally, the manuscript is well written. The introduction is well organized. It is easy to follow the idea of the manuscript as well as the materials and methods section.

In my opinion, there is missing or not intensively highlighted factor such as:

- for how long and how were the dose were applied for rats i.e. 1 dose each day or 1 or several doses per day?

- what kind of compounds were identified that can affect the abortion effects.

A summary is needed.

line 76 MP - add full name where occurred for the first time in the main manuscript body.

materials and methods section should be before the conclusions

Please check the grammar.

Author Response

Rebuttal Letter

The authors appreciate the reviewer’s efforts in revising the manuscript entitled: “Mentha pulegium L. (pennyroyal, Lamiaceae) extracts impose abortion or fetal-mediated toxicity in pregnant rats; evidenced by the modulation of pregnancy hormones, MiR-520, MiR-146a, TIMP-1 and MMP-9 protein expressions, inflammatory state, certain related signaling pathways, and metabolite profiling via UPLC-ESI-TOF-MS”. Please find below our responses to the reviewer’s comments.

Reviewer #3

Comment

Response

1.     - for how long and how were the dose were applied for rats i.e. 1 dose each day or 1 or several doses per day?

The current work was conducted at two parts to reach the goal of this study. The first one was the dose selection part which is clearly illustrated in the material and method part  starting from line 658 page 22

Besides, the dosing regimens in standard or MP were also mentioned in detail on page 22 and 23; line 680-686. 

On the fifteenth day of gestation, female rats with confirmed pregnancies were randomly divided into five groups as follows: Group I: pregnant rats serving as positive control group. Group II: rats were treated by misoprostol (100 ug/kg; once per day orally for three days) to induce abortion and served as standard control group. This dose was selected after a pilot study (data not included), guided by previous studies used misoprostol alone in abortion induction [13, 100]. Group III, IV, and V received 125, 250, and 500 mg/kg of MP extract, respectively (once per day for three days starting on day 15th)

2.     - what kind of compounds were identified that can affect the abortion effects.

The findings of the study revealed that MP extract and all its fractions exerted varying effects causing pregnancy jeopardy and this action was not confined to one fraction only or a specific class solely, but rather a synergistic effect of the secondary metabolites of MP.

(a summary was added in the discussion section lines 631-645) which states:

It is worth mentioning that all MP fractions imposed fetal morphologic malformations if not complete abortion. In brief, all the fractions presented a significant reduction in fetal weight was detected especially with the But. and rem. Aq. fractions in addition to the highest intrauterine growth retardation (IUGR). Both fractions also caused elevation in the levels of estradiol with a decline in serum progesterone levels as compared to healthy pregnant ones and those treated with MP extract. The rem. Aq. fraction showed the highest increase in MiR-520 and TNF-α values, compared to MP, whereas MecH fraction exhibited the highest effect on TIMP-1 and MMP-9 expressions. EtOAc treated rats showed depressive behaviors contrarily to Rem. aq. treated ones which presented a state of anxiety and stress. Misoprostol treated groups showed similar, or even more pronounced behavioral state, to EtOAc treated group. MP extract and its fractions showed a reduced GFAP content, and misoprostol presented the least GFAP contents. Misoprostol, MP extract, and EtOAc fraction increased cortical content of 5-HT. All the treatments, except for the But. fraction, decreased the cortical contents of BDNF as compared to those of positive control reflecting neurogenesis inhibition.

3.     A summary is needed.

4.      line 76 MP - add full name where occurred for the first time in the main manuscript body.

Done

5.      materials and methods section should be before the conclusions

It has been placed after the conclusion section in accordance with the template provided by the journal

6.      Please check the grammar.

English language has been checked and changes are highlighted throughout the manuscript.

Reviewer 4 Report

The paper entitled “Mentha pulegium L. (pennyroyal, Lamiaceae) extracts impose  abortion or fetal-mediated toxicity in pregnant rats; evidenced by the modulation of pregnancy hormones, MiR-520, MiR-146a, TIMP-1 and MMP-9 protein expressions, inflammatory state, certain related signaling pathways, and metabolite profiling via UPLC-ESI-TOF-MS” submitted to Toxins journal aimed to in vivo studies based on rats model. The study are based on mentha extracts which is commonly used by pregnant women and considered as safe natural herbal product. The paper is very interesting and informative. Please consider the detail review :

  1. Abstract: Informative but too long. The abstract should be a total of about 200 words maximum.
  2. Introduction: consistent and informative.
  3. Methods: authors decided to use in vivo assay along with ex vivo examinations which are adequate for the aim of the study. The methodology is well described and facilitate repetition of the experiments.
  4. Results: presented in form of tables and figures. Authors presented all necessary study results which are well described. Tables should include ±SD information. Figure 3. and 4. should be improved because the quality is weak what causes problems with analysis.
  5. Discussion: adequate to obtained study results. References are up-to-date and adequate.
  6. Conclusions: informative and consistent.

Author Response

Rebuttal Letter

The authors appreciate the reviewer’s efforts for revising the manuscript entitled: “Mentha pulegium L. (pennyroyal, Lamiaceae) extracts impose abortion or fetal-mediated toxicity in pregnant rats; evidenced by the modulation of pregnancy hormones, MiR-520, MiR-146a, TIMP-1 and MMP-9 protein expressions, inflammatory state, certain related signaling pathways, and metabolite profiling via UPLC-ESI-TOF-MS”. Please find below our responses to the reviewer’s comments.

Reviewer #4

Comment

Response

1.Abstract: Informative but too long. The abstract should be a total of about 200 words maximum.

According to the authors’ guidelines the abstract should not exceed 200  words. This version of the abstract is 201 words which is comprehensive to all the results and  highly informative, and we were unable to delete any of them due to their importance

4. Results: presented in form of tables and figures. Authors presented all necessary study results which are well described. Tables should include ±SD information. Figure 3. and 4. should be improved because the quality is weak what causes problems with analysis.

-Concerning the statistical analysis table (1) was analyzed and the description of the analysis was included while table (2) no statistical analysis was done so no descriptive caption was included 

-As per the reviewer’s comment Figures 3. and 4. were converted to higher resolution (300 dpi) 

Round 2

Reviewer 1 Report

Thank you for the revision. I think you have to rewrite the abstract (background of plant use, need of project, methodology,results, and conclusion) using 200 words.

Author Response

The abstract has been updated according to the reviewer's comment. A background about the plant use was added, and the total number of words are within limit of 200 words.

the updated abstract is:

Pregnant women usually turn to natural products to relieve pregnancy-related ailments which might pose health risks. Mentha pulegium L. (MP, Lamiaceae) is a common insect repellent, and the present work validates its abortifacient capacity, targeting morphological anomalies, biological, and behavioral consequences compared to misoprostol. The study also includes untargeted metabolite profiling of MP extract and fractions thereof viz. methylene chloride (MecH), ethyl acetate (EtOAc), butanol (But), and the remaining liquor (Rem. Aq.) by UPLC-ESI-MS-TOF, to unravel the constituents provoking abortion. Administration of MP extract/fractions, for three days starting from day 15th of gestation, affected fetal development by perturbing the uterine and placental tissues or even caused pregnancy termination. These pathologies also entailed biochemical findings where they decreased progesterone and increased estradiol serum levels, modulated placental gene expressions of both MiR-(146a & 520), decreased uterine MMP-9, and up-regulated TIMP-1 protein expression, and empathized inflammatory responses (TNF-α, IL-1β). Besides, these potentials affected the brain's GFAP, BDNF, and 5-HT contents and some of the behavioral parameters escorted by the open field test.  All these incidences were also perceived in the misoprostol-treated group. A total of 128 metabolites were identified in the alcoholic extract of MP, including hydroxycinnamates, flavonoid conjugates, quinones, iridoids, and terpenes.